# Federated Causal Structure Learning with Non-identical Variable Sets

Yunxia Wang [1]  Fuyuan Cao [1]  Kui Yu [2]  Jiye Liang [1]

## Abstract

Federated causal structure learning aims to infer causal relationships from data stored on individual clients, with privacy concerns. Most existing methods assume identical variable sets across clients and present federated strategies for aggregating local updates. However, in practice, clients often observe overlapping but non-identical variable sets, and non-overlapping variables may introduce spurious dependencies. Moreover, existing strategies typically reflect only the overall quality of local graphs, ignoring the varying importance of relationships within each graph. In this paper, we study federated causal structure learning with non-identical variable sets, aiming to design an effective strategy for aggregating "*correct*" and "*good*" causal relationships across distributed datasets. Specifically, we first develop theories for detecting spurious dependencies, examining whether the learned causal and non-causal relationships are "*correct*" or not. Furthermore, we define stable relationships as those that are both "*correct*" and "*good*" across multiple graphs, and finally design a two-level priority selection strategy for aggregating local updates, obtaining a global causal graph over the integrated variables. Experimental results on synthetic, benchmark and real-world datasets demonstrate the effectiveness of our proposed method.

## 1. Introduction

Causal discovery holds significant roles in various scientific fields, including biology, epidemiology, medicine, and economics (Replogle et al., 2022; Cai et al., 2023; Squires et al., 2023; Feuerriegel et al., 2024; Li et al., 2024a). Unveiling the causal relationships between random variables from data

presents a challenging research task (Hernn & Robins, 2020; Wang et al., 2022; Zhu et al., 2024). With the rapid growth of data volume, data owners gradually refuse to share their personalized data to avoid privacy leakage (Cramer et al., 2015; Yang et al., 2019; Li et al., 2020), making the inference of causal graphs from centralized data troublesome. Consequently, in recent years, federated causal structure learning (CSL) has been increasing attentions in uncovering the causal relationships between variables from decentralized data, with a data-privacy protecting way.

Existing federated CSL methods are usually classified into constraint-based (Li et al., 2024b; Huang et al., 2023; Guo et al., 2024), score-based (Mian et al., 2023; Ye et al., 2024), and continuous-optimization-based (Abyaneh et al., 2022; Ng & Zhang, 2022; Gao et al., 2023; Chengbo & Kai, 2024) methods. Almost all of them focus on discovering causal relations among a set of identical variables, without considering latent variables underlying the data. Additionally, the federated strategies presented by these methods typically reflect only the overall quality of locally discovered causal graphs in every iteration, ignoring the varying importance of relationships between variables within each graph. For instance, PERI (Mian et al., 2023) learned the final global causal graph that minimizes the worst-case regret with respect to the locally discovered causal networks, using distributed min-max regret optimization. FedCSL (Guo et al., 2024) first estimated the sample size of each client and then assigned sample-size-based weights for the local results. Notears-ADMM (Ng & Zhang, 2022) relied on the ADMM (Boyd et al., 2011) optimization, FedDAG (Gao et al., 2023) employed the FedAvg (McMahan et al., 2017) technique, and FedCDH (Li et al., 2024b) constructed the summary statistics, all of which exchange statistics representing the whole local graphs.

In practice, often due to privacy, ethical, financial, and practical concerns, the variables observed by different clients are not entirely identical but partially overlapping (Tillman & Spirtes, 2011), and the data collected by clients might be suitable for uncovering the relationships between different variables. For example, multiple hospitals, due to variations in professional specialties, medical resources, and technical proficiency, measure non-identical indicators and are suited for diagnosing different types of diseases. Therefore, it is essential to study federated causal structure learning in

[1]School of Computer and Information Technology, Shanxi University, Taiyuan, China [2]School of Computer Science and Information Engineering, Hefei University of Technology, Hefei, China. Correspondence to: Fuyuan Cao <cfy@sxu.edu.cn>.

*Proceedings of the 42nd International Conference on Machine Learning*, Vancouver, Canada. PMLR 267, 2025. Copyright 2025 by the author(s).

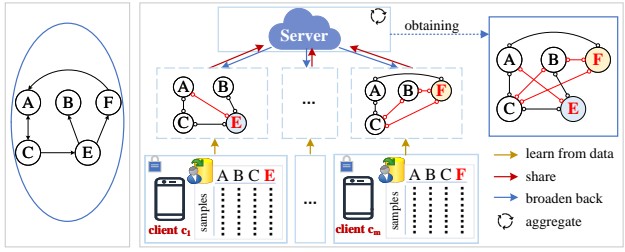

(a) GroundTruth    (b) The learned causal graph over the set of integrated variables

*Figure 1.* A toy example to illustrate spurious dependencies caused by non-overlapping variable pairs. Here, $E$ and $F$ are not observed together by any client, so their relationship is unknown and dependencies between several variables adjacent to them are spurious, such as $A\circ\!\!-\!\!\circ E$ in client $c_1$, $C\circ\!\!-\!\!\circ F$ in client $c_m$, and others.

the setting of non-identical variable sets, with an effective federated strategy.

However, designing such a federated strategy presents two main challenges. The first arises from the presence of non-overlapping variable pairs—those that are never observed together by any client—which may lead to spurious dependencies, as illustrated in Figure 1. Since no joint observations are available for these non-overlapping pairs, their relationships cannot be directly inferred from the input data. Moreover, the dependencies among overlapping pairs may be indirectly distort by non-overlapping ones: either variable in a non-overlapping pair may act as an unobserved confounder or mediator for the other and its adjacent variables, thereby introducing spurious dependencies. The second challenge is how to evaluate the importance of causal relationships within each locally discovered causal graph and then aggregate the "*good*" ones across iterations. While some distributed causal discovery methods take into account non-identical variable sets (Cao et al., 2024; Huang et al., 2020; Triantafillou & Tsamardinos, 2015; Tillman et al., 2008; Tillman & Spirtes, 2011), they ignore the issues mentioned above and typically allow raw data to be shared.

To address the above key challenges, we propose a novel Federated Causal Discovery algorithm for Non-identical Variable sets, called FedCDnv, aiming to design an effective federated strategy that aggregates both "*correct*" and "*good*" relationships during collaboratively learning, while preserving data privacy. The strategy involve two factors. One is that detecting whether the learned relationships are "*correct*" or not. For this, we develop theories to first determine whether the relationship between each non-overlapping variable pair is definitively non-causal and then examine which dependencies among overlapping variables are affected. The other is that analyzing which relationships are "*good*" ones. Here, we consider the stability of the process for learning causal relationships, with the perspective that setting an interval of significance levels for conditional independence (CI) tests, and regarding relationships that remain stable CI

tests as "*good*" ones. Based on the above considerations, we define stable causal and non-causal relationships as those that are both "*correct*" and "*good*" across multiple graphs. And then we design a two-level priority selection strategy (*TPSS*) in the context of non-identical variable sets, bridging local learning on clients with global aggregation on the server. Extensive experiments on synthetic, benchmark, and real-world data validate the effectiveness of FedCDnv.

## 2. Formal Preliminaries

When the causal sufficiency assumption (Spirtes et al., 2000) is violated, the underlying system is typically modeled using Semi-Markov Causal Models (SMCMs) (Tian & Pearl, 2002) or Maximal Ancestral Graphs (MAGs) (Richardson & Spirtes, 2002), rather than Directed Acyclic Graphs (DAGs) (Spirtes et al., 2000). We use $\mathcal{V} = \{O, \mathcal{L}, \mathcal{S}\}$ to denote a set of all variables describing the system, where $O$, $\mathcal{L}$, and $\mathcal{S}$ represent disjoint sets of observed, latent, and selection variables, respectively. In this work, we use MAGs to model the conditional independence relations among $O$, denoted by $\mathcal{M} = (O, E)$, where $E \subseteq O \times O$ is the set of edges including directed edges ($\rightarrow$), bi-directed edges ($\leftrightarrow$), and undirected edges ($-$). For $X_i, X_j \in O$, $X_i \rightarrow X_j$ denotes that $X_i$ is a direct cause of $X_j$, $X_i \leftrightarrow X_j$ denotes that the presence of latent confounders between $X_i$ and $X_j$, and $X_i - X_j$ indicates the presence of selection bias between them. MAGs with the exact set of conditional independencies are Markov equivalent, and the complete set is represented as a Partial Ancestral Graph (PAG). In a PAG, endpoints that can be either arrowheads ($>$) or tails ($-$) in different MAGs are denoted with a circle ($\circ$). In this work, we assume the presence of latent confounders but no selection bias.

Consider that there are $m$ clients in total, denoted by $C = \{c_1, \cdots, c_m\}$, and one central server, denoted by $S$. Each client has its own local dataset $D_k$ and observed variables $O_k$ ($k \in \{1, \cdots, m\}$). Let $n_k$ and $d_k$ denote the number of samples and variables in client $c_k$, respectively. The union of observed variables across $m$ clients is denoted by the set $O = \bigcup_{k=1}^{m} O_k$, with $d = |O|$ being the total number of variables. Since the variable sets across clients are non-identical, we categorize latent variables behind data into two types: *absolute latent variables* and *relative latent variables*. *Absolute latent variables* refer to the aforementioned $\mathcal{L}$, while *relative latent variables* exist in each client $c_k$, represented as $L_k = O \backslash O_k$. In addition, we use *non-overlapping variable pairs* to describe pairs of variables that are not observed simultaneously by any client, and similarly use *overlapping variable pairs* to represent pairs that are observed simultaneously by at least one client.

The core of our work is to identify causal relationships on the integrated variable set $O$ within the context of federated learning, using individual datasets with latent variables.

The assumptions required for this paper are presented as Assumptions 2.1, 2.2 and 2.3.

**Assumption 2.1.** Assume that $P_k$ is the joint probability distribution over $O_k$, and $\mathcal{M}_k$ is the MAG describing the causal relations among $O_k$ for the client $c_k$, then $P_k$ and $\mathcal{M}_k$ are assumed to be faithful to each other.

**Assumption 2.2.** All local datasets are uniformly sampled from the same causal DAG over $\mathcal{V} = \{O, \mathcal{L}, \mathcal{S}\}$, and the probability distribution of samples can differ across different clients. In addition, any two local datasets $D_i$, $D_j$ $(i, j \in \{1, \cdots, m\}, i \neq j)$ are not shared.

**Assumption 2.3.** Assume that the intersection of any two variable sets $O_i$ and $O_j$, observed by client $c_i$ and $c_j$ $(i \neq j; i, j \in \{1, \cdots, m\})$, is not empty, i.e., $O_i \cap O_j \neq \emptyset$.

## 3. The Proposed FedCDnv Algorithm

In this section, we present FedCDnv, which consists of three key submodules. The first submodule focuses on developing theories to detect whether the relationship between each variable pair, both non-overlapping and overlapping ones, is definite (i.e., "*correct*") causal or non-causal relationship. Based on the constructed theories, the second submodule aims to design a federated strategy suited for the context of non-identical variable sets. The third submodule presents the implementation details of FedCDnv.

### 3.1. Develop Theories

The setting of non-identical variable sets in federated causal discovery introduces two theoretical challenges. On the one hand, *non-overlapping variable pairs* may exist and their causal relationships cannot be theoretically identified due to the lack of their joint data. For this, our first focus is to determine whether there is a definite non-causal relationship between variables in each of these pairs. On the other hand, the presence of *non-overlapping variable pairs* may cause spurious dependencies. Our second focus is to examine whether the discovered causal relationships among *overlapping variable pairs* are spurious.

Assume that $G_k = (O_k, E_k)$ is a local causal graph learned from the dataset in client $c_k$ $(k \in \{1, \cdots, m\})$ with an oracle of conditional independence tests, where $O_k$ represents the variable set observed by $c_k$ and $E_k \subseteq O_k \times O_k$ represents the set of learned edges (including $\rightarrow$, $\leftrightarrow$, $\circ\!\!\rightarrow$, $\circ\!\!-\!\!\circ$). Let $G = (O, E)$ be an integrated graph over $O$, obtained by aggregating adjacencies and arrowheads (one of the orientations) from all local graphs (i.e., $O = \bigcup_{k=1}^{m} O_k$, $E = \bigcup_{k=1}^{m} E_k$). We first classify any two variables $X, Y \in O$ into one of the two following categories and then propose the following theories.

I $\langle X, Y \rangle$ is a *non-overlapping variable pair*, i.e., $X, Y$ are not observed simultaneously by any client;

II $\langle X, Y \rangle$ is an *overlapping variable pair* and they are observed simultaneously by at least one client.

For Category I, due to the absence of joint data for non-overlapping pairs, their relationships are initially considered to be non-definitely non-causal ones, meaning it is unknown whether edges exist between variables of these pairs in the ground truth. To examine which of these relationships are definite non-causal, we introduce Theorem 3.1 as follows.

**Theorem 3.1.** *Let $\langle X, Y \rangle$ be a non-overlapping variable pair $(X, Y \in O)$ and $\mathcal{A}_X^G$ be the set of variables adjacent to $X$ in $G$. If there exists $\mathcal{A}_X^G \cap \mathcal{A}_Y^G = \emptyset$ & $\mathcal{A}_X^G \neq \emptyset$ & $\mathcal{A}_Y^G \neq \emptyset$, and for each $Y' \in \mathcal{A}_Y^G$, $\langle X, Y' \rangle$ is not a non-overlapping variable pair (and vice versa), then the relationship between $X$ and $Y$ is considered definitively non-causal.*

Theorem 3.1 utilizes the properties of adjacency sets of variables in non-overlapping variable pairs to examine whether the relationship between them is definitively non-causal.

For Category II, we initially assume the causal and non-causal relationships among overlapping pairs are definite, or "*correct*". However, the presence of non-overlapping pairs may introduce spurious dependencies among overlapping pairs. For this, Lemmas 3.2 and 3.3 are introduced to detect which of these learned relationships are non-definite.

**Lemma 3.2.** *Assume that $X$ and $Y$ $(X, Y \in O)$ are observed simultaneously by at least one client and $\mathbf{Z}_n$ represents a union set of variables that appear in non-overlapping variable pairs with non-definite, non-causal relationships. If $X \in \mathbf{Z}_n$ or $Y \in \mathbf{Z}_n$, then the causal relationship between $X$ and $Y$ is considered to be non-definite.*

In Lemma 3.2, if a non-overlapping variable pair cannot be determined to be definitively non-causal by Theorem 3.1, then the causal relationships between each variable in the pair and its adjacent variables, are considered non-definite.

**Lemma 3.3.** *Assume that $X$ and $Y$ $(X, Y \in O)$ are observed simultaneously by at least one client and $\mathcal{A}_X^G$ is a set of variables adjacent to $X$ in $G$. If there exists a non-overlapping variable pair $\langle A, B \rangle$ such that $\{A, B\} \subseteq \mathcal{A}_X^G \cup \mathcal{A}_Y^G$, then the causal relationship between $X$ and $Y$ is considered to be non-definite.*

In Lemma 3.3, the adjacency between $X$ and $Y$ identified by the raw data may arise due to incomplete or incorrect separating sets. In other words, the ideal separating sets—those that should render $X$ and $Y$ independent—may contain pairs of variables that cannot be observed together by any client. Consequently, no valid separating set exists that can fully ensure the independence of $X$ and $Y$.

### 3.2. Design the Federated Strategy

In this subsection, we design a federated strategy—two-level priority selection strategy (*TPSS*), aiming to aggregate

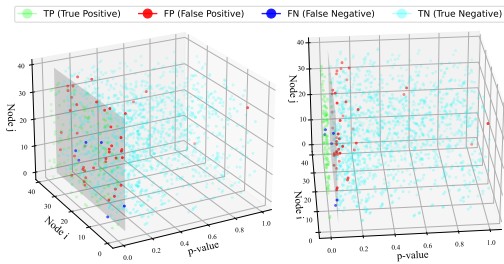

*Figure 2.* An example of p-value distributions between 40 nodes from different perspectives, with the left subplot showing a frontal view and the right one displaying an adjusted angle. The gray plane represents the significance level $\alpha$ (p-value) = 0.05, with false positives and false negatives concentrated around $\alpha$.

both "*correct*" and "*good*" relationships across different clients. It contains two steps as follows.

***Step1: Define stable relationships.*** To represent both "*correct*" and "*good*" relationships across multiple local causal graphs, we introduce the concept of *stable relationships*.

Let $G_k = (O_k, E_k)$ denote a causal graph learned from the dataset of client $c_k$ ($k \in \{1, \cdots, m\}$), and let $\{p_{ij}^{c_k}\}$ be the set of p-values obtained by conducting conditional independence (CI) tests on $X_i$ and $X_j$ ($X_i, X_j \in O_k$), conditioned on the separating sets in client $c_k$. Given an interval of significant level $[\alpha - \theta_1, \alpha + \theta_2]$ ($0 < \theta_1 < \alpha$, $\alpha < \theta_2 < 1 - \alpha$, and $\alpha$ is the significant level), we define the stable causal and non-causal relationships in $G_k$ as follows.

**Definition 3.4. stable causal relationship**. In a causal graph $G_k$, if the relationship between $X_i$ and $X_j$ is determined to be definite causal, and the maximum of p-values $\bar{p}_{ij}^{c_k}$ for $X_i \not\perp\!\!\!\perp_{D_k} X_j | \mathbf{Z}$ ($\forall \mathbf{Z} \subseteq O_k \backslash \{X_i, X_j\}$) is lower than $\alpha - \theta_1$, we call the relationship between $X_i$ and $X_j$ a stable causal relationship in $G_k$.

**Definition 3.5. stable non-causal relationship**. In a causal graph $G_k$, if the relation between $X_i$ and $X_j$ is determined to be definite non-causal, and there exists a set $\mathbf{Z} \subseteq O_k \backslash \{X_i, X_j\}$ such that $X_i \perp\!\!\!\perp_{D_k} X_j | \mathbf{Z}$ with a corresponding p-value greater than $\alpha + \theta_2$, we call the relationship between $X_i$ and $X_j$ a stable non-causal relationship in $G_k$.

The definition of stable relationships involves two characteristics. First, a stable relationship must be consistent with those in the ground truth (i.e., it is a definite relationship). If the relation between $X_i$ and $X_j$ deviates from the ground truth, it will become unstable, thereby introducing errors. The second characteristic arises from an intuition that, in practice, when the p-value $p$ is very small (e.g., $p \leq 0.001$), the null hypothesis can be rejected, and when $p$ is very large (e.g., $p > 0.5$), there is insufficient evidence to reject the null hypothesis. Figure 2 illustrates the experimental distribution of false positives and false negatives relative to the confidence level $\alpha$, which shows that most of false positives and false negatives are concentrated around the confidence

level, which is within the range $[\alpha - \theta_1, \alpha + \theta_2]$. This provides empirical evidence supporting the intuition.

***Step2: Present the details of TPSS.*** *TPSS* contains two levels of priority. The first level is to identify the inconsistent causal relationships caused by the non-identical variable sets. And the second level aims to evaluate the varying importance of relationships between variables.

Assume that $\langle X_i, X_j \rangle$ ($X_i, X_j \in O$) is an *overlapping variable pair*. Let $\{G_{k'}\}$ denote the set of local graphs where $X_i$ and $X_j$ are observed together, corresponding to the set of clients $\{c_{k'}\}$. If $X_i$ and $X_j$ are adjacent in $\{G_{k'_a}\}$ but not adjacent in the remaining graphs $\{G_{k'}\} \setminus \{G_{k'_a}\}$ (denoted as $\{G_{k'_n}\}$), then *TPSS* begins to execute.

At the first level, we identify whether the inconsistent adjacencies between $X_i$ and $X_j$ in $\{G_{k'_a}\}$ and $\{G_{k'_n}\}$ are caused by non-identical variable sets across clients, and introduce Conditions 3.6 and 3.7 as follows.

**Condition 3.6.** *For variables $X_i$, $X_j$ with a bi-directed edge between them in client $c_{k'_a}$, if there exists a client $c_{k'_n}$ such that $k'_a \neq k'_n$, $X_i \leftarrow\!\!\circ Y \circ\!\!\rightarrow X_j \in E_{k'_n}$, and $Y \notin O_{k'_a}$, then $Y$ is an unobserved confounder for client $c_{k'_a}$.*

**Condition 3.7.** *For variables $X_i$, $X_j$, which take forms such as $X_i \circ\!\!-\!\!\circ X_j$, $X_i \circ\!\!-\!\!\rightarrow X_j$, or $X_i \leftarrow\!\!-\!\!\circ X_j$ in client $c_{k'_a}$, their inconsistent adjacencies arise from non-identical observed variable sets, under the following condition: for any $G_{k'_n} \in \{G_{k'_n}\}$, there exists a minimal separating set $\mathbf{Z} \subseteq O_{k'_n}$ such that $X_i \perp\!\!\!\perp X_j | \mathbf{Z}$ holds in $D_{k'_n}$, and for every $G_{k'_a} \in \{G_{k'_a}\}$, the variables in $\mathbf{Z}$ are never simultaneously observed—that is, there does not exist a subset of $O_{k'_a}$ that contains all variables in $\mathbf{Z}$, and $X_i, X_j \notin \mathbf{Z}$.*

If the first level of priority is not satisfied, then we suspect that the inconsistency of $X_i$ and $X_j$ in $\{G_{k'}\}$ is caused by inaccurate CI tests. Thus, at the second level of priority, we propose a way for evaluating varying importance of relationships between different variables. It involves two aspects: "*correct*" and "*good*" relationships for each client, which are represented as *stable relationships*; the impact of varying sample sizes across clients on these relationships.

Specifically, for variables $X_i$ and $X_j$, we use 1 to represent the importance of stable relationships between them, $w_{ij}^{c_{k'}}$ ($0 < w_{ij}^{c_{k'}} < 1$) to indicate the importance of non-stable relationships relative to stable ones, and $w_\tau^{c_{k'}}$ to account for the impact of the sample size of client $c_{k'}$, where $c_{k'} \in \{c_{k'}\}$. The adjacency of $X_i$ and $X_j$ is determined by comparing $val_{ij}$ with 0, where $val_{ij}$ is computed as follows.

$$val_{ij} = \sum_{c_{k'} \in \{c_{k'}\}} w_\tau^{c_{k'}} \times w_{ij}^{c_{k'_s}}, \qquad (1)$$

where $w_\tau^{c_{k'}} = \frac{n_{k'}}{n'}$ ($n_{k'}$ denotes the sample size of client $c_{k'}$, and $n' = \sum_{c_{k'} \in \{c_{k'}\}} n_{k'}$ is the total sample size across

clients $\{c_{k'}\}$), and $w_{ij}^{c_{k'_s}}$ takes the values: 1, -1, $w_{ij}^{c_{k'}}$, $-w_{ij}^{c_{k'}}$. Specifically, 1, -1 indicate the importance of stable causal and non-causal relationships, respectively, whereas $w_{ij}^{c_{k'}}$ and $-w_{ij}^{c_{k'}}$ reflect the importance of non-stable causal and non-causal relationships, respectively.

Here, $w_{ij}^{c_{k'}}$ is calculated adaptively using p-values. Specifically, when the p-value $p_{ij}^{c_{k'}}$ falls within $[\alpha - \theta_1, \alpha]$ or $(\alpha, \alpha + \theta_2]$, it is uniformly scaled to the interval $[0, 1]$ using (2).

$$
\hat{p}_{ij}^{c_{k'}} = \begin{cases} \frac{\alpha - p_{ij}^{c_{k'}}}{\theta_1} & \text{if } p_{ij}^{c_{k'}} \in [\alpha - \theta_1, \ \alpha] \\ \frac{p_{ij}^{c_{k'}} - \alpha}{\theta_2} & \text{if } p_{ij}^{c_{k'}} \in (\alpha, \ \alpha + \theta_2] \end{cases}, \qquad (2)
$$

where $\hat{p}_{ij}^{c_{k'}}$ denotes the scaled p-value of $X_i$ and $X_j$. Then $w_{ij}^{c_{k'}}$ is obtained using (3).

$$
w_{ij}^{c_{k'}} = \frac{\hat{p}_{ij}^{c_{k'}} \times |\{c_{k'}\}|}{\sum_{c_{k'_\rho} \in \{c_{k'_\rho}\}} \hat{p}_{ij}^{c_{k'_\rho}} + \sum_{c_{k'_\varrho} \in \{c_{k'_\varrho}\}} 1}, \qquad (3)
$$

where $\{c_{k'_\rho}\}$ and $\{c_{k'_\varrho}\}$ denote the sets of clients that identify the relationship between $X_i$ and $X_j$ as non-stable and stable, respectively. If this relationship is identified as non-stable but the corresponding p-value does not fall within the interval $[\alpha - \theta_1, \alpha + \theta_2]$, then $\hat{p}_{ij}^{c_{k'}} = p_{ij}^{c_{k'}}$.

### 3.3. Present the FedCDnv Algorithm

FedCDnv outputs two graphs: FedG, a global causal graph over $O$; and FeddG, a subgraph of FedG that contains only the definite causal and non-causal relationships. The latter is derived by further applying proposed theories to extract definite relations from FedG.

#### 3.3.1. Obtaining FedG

The algorithm description of FedCDnv is presented in Algorithm 1, which involves two rounds of interaction between clients and the central server.

FedCDnv first applies an existing causal discovery method (e.g., FCI (Spirtes et al., 2000) or RFCI (Colombo et al., 2012)) to the local dataset of each client, obtaining a causal graph over $O_k$, denoted by $\text{Pag}_0^{c_k}$, and the corresponding set of p-values, denoted by $\text{p}^{c_k}$. Each client then shares its local graph $\text{Pag}_0^{c_k}$ and sample size $n_k$ with the central server $S$. At the server, the adjacencies and orientations from all local graphs $\{\text{Pag}_0^{c_k}\}_{k=1}^m$ are integrated into a global graph $G_0 = (O, E_0)$. Specifically, for adjacencies, a variable pair is considered adjacent if any client reports the adjacency between them. While for orientations (there are three types of orientations: circle 'o', tail '−', and arrow '>'), only the identified arrows are included in the global graph. For example, for $X_i \circ\!\!-\!\!\circ X_j$, $X_i \circ\!\!\rightarrow X_j$, and $X_i \rightarrow X_j$, the final relationship between $X_i$ and $X_j$ will be $X_i \circ\!\!\rightarrow X_j$.

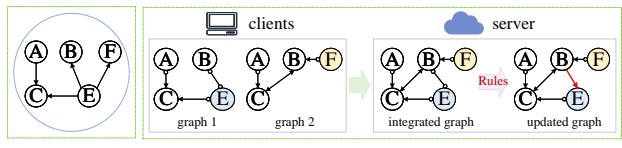

(a) GroundTruth      (b) The integrated causal graph during an interaction

*Figure 3.* An example for explaining the cases that rules described by Zhang (Zhang, 2008) are not applicable to the integrated graph $G_0$. Based on the rules, $B$ and $E$ are oriented into $B \rightarrow E$ in $G_0$, which is not consistent with $E \rightarrow B$ in GroundTruth.

Obtaining $G_0$ serves two main purposes. First, it enables the determination of whether the relationship between each pair of variables is definite, upon which the status matrix $stat$ is constructed (as explained below). Second, it allows for updating each local graph $\text{Pag}_0^{c_k}$, as the orientation rules proposed by Zhang (Zhang, 2008) are not directly applicable to $G_0$, as illustrated in Figure 3.

The state matrix $stat$ takes values from the set $\{-2, -1, 1, 2\}$, each encoding a different type of relationship between two variables. Let $G = (O, E)$ denote the integrated graph and $\mathcal{M}$ the ground truth. $stat(i, j) = -2$ represents a definite non-causal relationship between $X_i$ and $X_j$, indicating that they are not adjacent in either $G$ or $\mathcal{M}$. $stat(i, j) = -1$ denotes a non-definite non-causal relationship between $X_i$ and $X_j$, meaning that they are not adjacent in $G$, but their relation in $\mathcal{M}$ is unknown—they may or may not be adjacent. $stat(i, j) = 2$ represents a definite causal relationship between $X_i$ and $X_j$, meaning that they are adjacent in both $G$ and $\mathcal{M}$. Lastly, $stat(i, j) = 1$ denotes a non-definite causal relationship between $X_i$ and $X_j$, where they are adjacent in $G$, but it is unknown whether they are adjacent in $\mathcal{M}$.

Next, $G_0$ is broadcast back to all clients, prompting each client to update its local causal graph, denoted as $\text{Pag}_u^{c_k}$. And in each client, p-values falling within the interval $[\alpha - \theta_1, \alpha + \theta_2]$ are scaled to $[0, 1]$ using (2), and then multiplied by -1, resulting in $\tilde{\text{p}}^{c_k}$. It is worth noting that scaling only the p-values within $[\alpha - \theta_1, \alpha + \theta_2]$ introduces a level of uncertainty that may help obscure precise statistical information, thereby enhancing privacy protection to some extent. The detailed algorithm is shown in Algorithm 2. Finally, $\text{Pag}_u^{c_k}$ and $\tilde{\text{p}}^{c_k}$ of each client are shared to $S$ again. In $S$, FedCDnv applies the proposed *TPSS* to aggregate adjacencies. For orientations, only the identified arrows are incorporated in the global graph. The detailed algorithm description is shown as Algorithm 3.

#### 3.3.2. Obtaining FeddG

We apply the proposed theories in Section 3.1 to extract definite relationships from FedG, obtaining FeddG.

For each variable pair $\langle X_i, X_j \rangle$, FedCDnv initially divides it into one of two categories. For Category I, Theorem 3.1

**Algorithm 1** FedCDnv

1: **Input:** $\{D_1, \cdots, D_m\}$, $\alpha$, $\theta_1$, $\theta_2$, FCI parameters *para*
2: **Output:** FedG, FeddG
3: **for** each client $c_k \in \{c_1, \cdots, c_m\}$ **do**
4:    ($\text{Pag}_0^{c_k}$, $\text{p}^{c_k}$) = C-InitPag($D_k$, $\alpha$, *para*) /*using FCI*/
5:    $n_k \leftarrow$ obtain the sample size of client $c_k$
6:    send $\text{Pag}_0^{c_k}$ and $n_k$ to the central server $S$
7: **end for**
8: $G_0 \leftarrow$ in $S$, transfer adjacencies and arrowheads of all graphs ($\{\text{Pag}_0^{c_k}\}_{k=1}^m$) into $G_0$
9: $stat \leftarrow$ in $S$, apply Theorem 3.1, Lemmas 3.2, and 3.3 to detect whether the learned relation between each variable pair is definite or not, obtaining the status matrix $stat$
10: broadcast $G_0$ back to all clients
11: **for** each client $c_k \in \{c_1, \cdots, c_m\}$ **do**
12:    $\text{Pag}_u^{c_k} \leftarrow$ C-Update($G_0$, $D_k$, $\text{Pag}_0^{c_k}$, $\alpha$)
13:    $\tilde{\text{p}}^{c_k} \leftarrow$ scaled p-values in $[\alpha - \theta_1, \alpha + \theta_2]$ using (2), multiplied by -1
14:    send $\text{Pag}_u^{c_k}$ and $\tilde{\text{p}}^{c_k}$ to the central server $S$
15: **end for**
16: FedG $\leftarrow$ S-FedG ($\{\text{Pag}_u^{c_k}\}$, $\{\tilde{\text{p}}^{c_k}\}$, $\{n_k\}$, $\alpha$, $\theta_1$, $\theta_2$, $stat$)
17: FeddG $\leftarrow$ according to obtained $stat$, extracting definite relations from FedG

---

**Algorithm 2** C-Update

1: **Input:** $G_0$, $D_k$, $\text{Pag}_0^{c_k}$, $\alpha$
2: **Output:** $\text{Pag}_u^{c_k}$
3: $O_k \leftarrow$ obtain the set of observed variables from $\text{Pag}_0^{c_k}$
4: **for** each v-structure $\mathbf{V} = \{X_i, Y, X_j\}$ in $G_0$, where $Y$ is the collider **do**
5:    **if** $\mathbf{V} \subseteq O_k$, $X_i, Y$ are adjacent and so do $X_j, Y$ in $\text{Pag}_0^{c_k}$ **then**
6:      orient $X_i \circ \!\!\rightarrow Y$ and $X_j \circ \!\!\rightarrow Y$ in $\text{Pag}_u^{c_k}$
7:    **end if**
8:    $\mathcal{A}_Y^{c_k} \leftarrow adj(Y; \text{Pag}_0^{c_k})$
9:    **for** each $B \in \mathcal{A}_Y^{c_k} \backslash \{X_i, X_j\}$ **do**
10:      **if** $\exists \mathbf{Z} \subseteq \mathcal{A}_Y^{c_k} \backslash \{B, X_i, X_j\}$ such that $X_i \perp\!\!\!\perp B|\{\mathbf{Z} \cup Y\}$ or $X_j \perp\!\!\!\perp B|\{\mathbf{Z} \cup Y\}$ **then**
11:        orient $Y \rightarrow B$ in $\text{Pag}_u^{c_k}$
12:      **end if**
13:    **end for**
14: **end for**

---

is applied directly. For Category II, Lemma 3.2 and Lemma 3.3 are called in sequence. All (non-)definite relationships are represented by different values in $stat$. FedCDnv extracts the relationships with $stat(i, j) = $ -2 and $stat(i, j) = 2$ from FedG as definite ones, obtaining FeddG.

### 3.4. Privacy and Costs Analysis

#### 3.4.1. PRIVACY ANALYSIS

To protect data privacy in the federated setting, FedCDnv avoids sharing raw data between clients and the server. Instead, structural information (i.e., local graphs $\text{Pag}_0^{c_k}$, $\text{Pag}_u^{c_k}$, and the global graph $G_0$) and statistical characteristics (i.e., $\tilde{\text{p}}^{c_k}$ and $n_k$) are exchanged. Notably, the p-values within the interval $[\alpha - \theta_1, \alpha + \theta_2]$ are first scaled using Eq. (2)

---

**Algorithm 3** S-FedG

1: **Input:** $\{\text{Pag}_u^{c_k}\}$, $\{\tilde{\text{p}}^{c_k}\}$, $\{n_k\}$, $\alpha$, $\theta_1$, $\theta_2$, $stat$
2: **Output:** FedG
3: $O \leftarrow$ the set of integrated variables
4: **for** each $X_i, X_j \in O$ **do**
5:    **if** Conditions 3.6 and 3.7 are satisfied **then**
6:      $X_i$ and $X_j$ are not adjacent in FedG
7:    **else**
8:      **for** each client $c_{k'} \in \{c_{k'}\}$, where $X_i$ and $X_j$ are jointly observed **do**
9:        compute $w_\tau^{c_{k'}}$ and $w_{ij_s}^{c_{k'}}$ based on $\text{Pag}_u^{c_{k'}}$, $\{n_k\}$, $stat$, and $\tilde{\text{p}}^{c_{k'}}$
10:      **end for**
11:      $val_{ij}$ is computed using (1)
12:      **if** $val_{ij} > 0$ **then**
13:        $X_i$ and $X_j$ are adjacent in FedG
14:      **end if**
15:    **end if**
16: **end for**
17: transfer arrowheads from each $\text{Pag}_u^{c_k} \in \{\text{Pag}_u^{c_k}\}$ to FedG

---

and then multiplied by -1, resulting in the transformed set of p-values, denoted by $\tilde{\text{p}}^{c_k}$. This transformation reduces the risk of leaking precise statistical characteristics, thereby enhancing privacy to a certain extent.

To further avoid data privacy leakage, several secure computation techniques can be considered in this paper. For example, secure multiparty computation (Cramer et al., 2015) is utilized to encrypt the exchanged graphs and sample sizes, enabling multiple clients to collectively compute a function over their inputs while keeping the data private. Alternatively, homomorphic encryption (Acar et al., 2018) facilitates the processing of encrypted data through complex mathematical operations without compromising the encryption. Furthermore, to prevent the leakage of semantic information related to variables, methods such as assigning unique identifiers to each variable can also be introduced. Also a certain client can be selected as a proxy server to avoid the leakage of the graph structure (Gao et al., 2023). Further privacy protection efforts will be explored in the future works.

#### 3.4.2. COMMUNICATION COSTS

Assume that the client $c_k$ observes $d_k$ variables, and the server $S$ integrates a total of $d$ variables ($d \geq d_k$, and "=" holding only when all $m$ clients observe identical variable sets). For each client $c_k$, two rounds of communication with the server are involved: one for sending the initially learned graph and another for sharing the updated local graph. In the first round, the client sends a $d_k \times d_k$ matrix ($\text{Pag}_0^{c_k}$), a value $n_k$, and receives a $d \times d$ matrix ($G_0$) from the server, which results in a communication cost of $O(d_k^2 + d^2 + 1)$. In the second round, the client sends the updated graphs $\text{Pag}_u^{c_k}$ and scaled statistical information $\tilde{\text{p}}^{c_k}$, also a $d_k \times d_k$ matrix, incurring a cost of $O(2d_k^2)$. Therefore, the total communication cost for a single client $c_k$ is $O(3d_k^2 + d^2 + 1)$,

and the total cost across all $m$ clients is $O(\sum_{k=1}^{m}(3d_k^2 + d^2 + 1))$. Since $d_k \leq d$, the overall communication cost is upper-bounded by $O(4md^2 + m)$, which can be further approximated as $O(md^2)$ in large-scale settings.

# 4. Experiments

In this section, we conduct extensive experiments to evaluate the effectiveness of the proposed FedCDnv. First, we describe the experimental setting in Section 4.1 (with further details provided in Appendix D.1). Then, in Section 4.2 and Appendices D.2-D.3, we compare FedCDnv with state-of-the-art methods from two perspectives: i) distributed CSL methods over non-identical variable sets, and ii) federated CSL methods over identical variable sets. Next, in Section 4.3, we verify the effectiveness of the definitely causal and non-causal relationships identified by FedCDnv, and we perform a sensitivity analysis of the stable relationships in Appendix D.4. Finally, in Appendix D.5, we evaluate the impact of the parameters listed in Table 4 on FedCDnv.

## 4.1. Experimental Settings

***Datasets***. (1) Synthetic data. The underlying DAGs are generated using the Erdös-Rényi (Erdös & Rényi, 1959) graph model with the graph size $n$. Due to the presence of *absolute latent variables*, the corresponding MAG's graph size is set to $d = n \times (1 - \lambda)$ $(0 < \lambda \leq 1)$. That is, the total number of observed variables integrated across all clients is $d$ $(|O| = d)$. For each client, the number of observed variables is set to $d_k = d \times \delta$, where $0 < \delta \leq 1$ denotes the ratio of client-specific observed variables to the total integrated variables. (2) Benchmark data. We use 6 networks with ranging from 20 to 111 variables: Child, Alarm, Insurance, Barley, Child3, and Alarm3. (3) Real-world data. We use "Sachs" (Sachs et al., 2005) to evaluate the performance of the methods, randomly selecting 9 out of 11 variables as those observed by each client.

***Baselines***. We conduct two types of experiments. (1) We compare our method with distributed CSL methods over non-identical variable sets, including FedCDnv-vote, C-DUIOV (Cao et al., 2024), CD-MiNi (Huang et al., 2020), and FCI-base. FedCDnv-vote is a variant of FedCDnv with the voting strategy. FCI-base is a baseline constructed by integrating causal graphs discovered by FCI on each client, using a sample-size-based federated strategy. (2) We compare FedCDnv with the state-of-the-art federated CSL methods in the setting of identical variable sets. Since FedCDnv still assumes the existence of *absolute latent variables*, we focus only on evaluating the performance of the learned skeleton. This comparison is intended to serve as a rough performance reference rather than a strict evaluating. The compared methods include FedCDnv-vote, Notears-ADMM (Ng & Zhang, 2022), FedPC (Huang et al., 2023), and Fed-

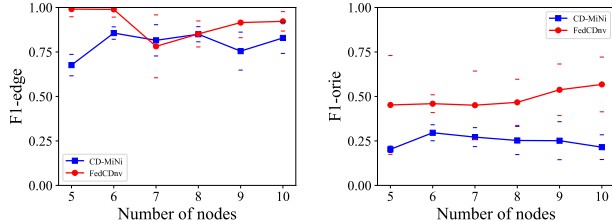

*Figure 4.* CDMiNi vs. FedCDnv

CSL (Guo et al., 2024).

***Metrics***. (1) We use precision, recall, and F1 score for returned edges and orientations, denoted as pre-edge, rec-edge, F1-edge, pre-orie, rec-orie, and F1-orie, to evaluate the returned causal graphs over non-identical variable sets. (2) We use pre-edge, rec-edge, and F1-edge to evaluate causal skeletons learned by the federated CSL methods with identical variable sets. (3) To test the accuracy of definite causal and non-causal relationships returned by FedCDnv, we use the false discovery rate (FDR) for them, denoted as FDR-C, FDR-dC, FDR-nC, FDR-dnC.

***Parameters***. For each invocation of FedCDnv, the default settings for each problem instance (set of datasets) are generated using the values shown in Table 4 of Appendix D.1. The default parameters are as follows: $G^2$ tests of conditional independence, the number of clients $|C| = 6$, the significance level $\alpha = 0.05$, $\theta_1 = 0.049$, $\theta_2 = 0.45$, $\lambda = 0.15\%$, $\delta = 0.85\%$, the sample size $n_k \in [100, ns]$ with $ns = 2000$.

## 4.2. Comparison of FedCDnv, FedCDnv-vote, CDUIOV, CD-MiNi, and FCI-base

### 4.2.1. FEDCDNV VS. CD-MINI

Since CD-MiNi is suitable for small-scale networks, we compare it with our method using synthetic data generated from networks containing 5 to 10 nodes, and the experimental results are shown in Figure 4. The vertical error bars (i.e., whiskers) indicate the range of one standard deviation above and below the mean. This design is chosen for clarity in comparing multiple methods.

It can be observed that FedCDnv exhibits relatively superior performance compared with CD-MiNi. Specifically, for F1-edge, while FedCDnv is slightly inferior to CD-MiNi in the network with 7 nodes and both are comparable in the network with 8 nodes, FedCDnv significantly outperforms CD-MiNi in remaining networks, i.e., networks with 5, 6, 9 and 10 nodes. As for the F1-orie metric, FedCDnv performs better than CD-MiNi across all six cases displayed. This difference may be due to FedCDnv considering the inconsistent causal relationships caused by *relative latent variables*, which could remove certain spurious edges. Additionally, FedCDnv effectively addresses specific inconsistent adjacen-

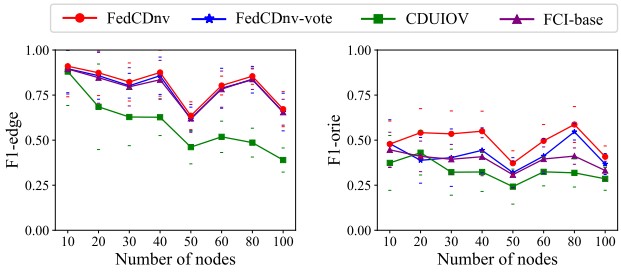

Figure 5. Experimental results on synthetic data.

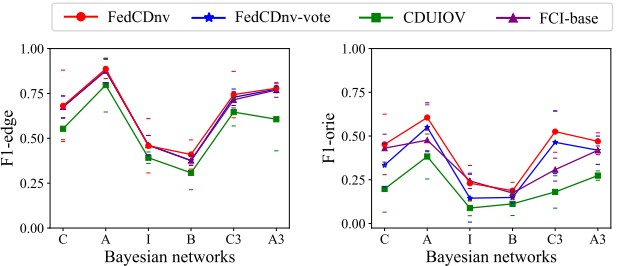

Figure 6. Experimental results on benchmark data.

cies by further considering the importance of relationships, thereby enhancing the reliability of causal discovery.

### 4.2.2. FEDCDNV VS. FEDCDNV-VOTE VS. CDUIOV VS. FCI-BASE

***Synthetic Data***. Figure 5 presents the experimental results for F1-edge and F1-orie of FedCDnv, FedCDnv-vote, CDUIOV, and FCI-base on synthetic data. It is observed that FedCDnv exhibits the highest F1-edge and F1-orie across almost all networks, with its orientations being nearly 10% better than those of the compared methods.

Specifically, for F1-edge, we observe that FedCDnv outperforms all other algorithms, followed by FedCDnv-vote and FCI-base, where FedCDnv-vote is slightly higher than FCI-base but not significantly, and CDUIOV performs the worst. For F1-orie, For F1-orie, FedCDnv again achieves the best performance across nearly all networks, except for the 10-node network. FedCDnv-vote's F1-orie is slightly higher than that of FCI-base in most networks, except in the 20-node network. CDUIOV shows a lower F1-orie than all other methods. This inferior performance of CDUIOV on both F1-edge and F1-orie is likely due to its framework being specifically designed for intervention data and its sensitivity to the sample sizes of local datasets. In addition, compared with FedCDnv-vote, the results indicate that the federated strategy proposed by FedCDnv is effective, supporting the consideration of both "*correct*" and "*good*" relationships. Finally, compared with other distributed methods, the superior performance of FedCDnv on synthetic data further demonstrates its effectiveness.

***Benchmark Data***. Figure 6 presents the results of FedCDnv, FedCDnv-vote, CDUIOV, and FCI-base on the benchmark networks. Here, six benchmarks are as the labels of the horizontal axis: 'C', 'A', 'I', 'B', 'C3', 'A3'. It is observed that FedCDnv demonstrates relatively superior performance compared with the baseline algorithms. While FedCDnv's edge-learning performance is only slightly better than existing methods, it shows a significant advantage in orientation identification across most networks.

Specifically, for F1-edge, FedCDnv achieves optimal perfor-

mance in the "Barley" and "Child3" networks, and performs slightly better than other methods in the remaining four networks. In contrast, the performance of FedCDnv-vote and FCI-base is comparable, but both are slightly inferior to FedCDnv. For F1-orie, FedCDnv surpasses the compared algorithms by a margin of 8%-12% in four networks. However, on "Insurance" and "Barley", its performance advantage is less pronounced. Furthermore, FedCDnv consistently outperforms FedCDnv-vote across all networks, demonstrating the effectiveness of the proposed *TPSS* on benchmark datasets. Between FedCDnv-vote and FCI-base, their performances are relatively balanced; for example, FedCDnv-vote outperforms in networks like "Alarm", whereas FCI-base performs better in networks like "Insurance". In contrast, CDUIOV exhibits suboptimal performance in both edge and orientation identification. The reason for this occurrence is similar to what was discussed for synthetic data above.

***Real-world Data***. Table 1 presents the experimental results of FedCDnv, FedCDnv-vote, CDUIOV, and FCI-base on "Sachs". The results indicate that FedCDnv demonstrates a significant advantage in identifying adjacencies between variables, as reflected by its superior performance in terms of pre-edge, rec-edge, and particularly F1-edge. Notably, FedCDnv achieves nearly a 10% improvement in F1-edge compared to the other methods. This performance gain can be attributed to FedCDnv's ability to more effectively handle the heterogeneity and partial overlap in variable sets across clients. By designing two levels of *TPSS*, FedCDnv considers the impact of the presence of possible confounders and the varying importance of relationships, reducing spurious dependencies and enhancing the accuracy of learned edges. In terms of learned orientations, while FedCDnv does not achieve the highest precision or recall individually, it attains the best F1-orie. This may be due to the better performance of the edges identified by FedCDnv, which results in a relatively high rec-orie. It is also worth noting that while CDUIOV achieves the highest F1-orie value, this comes at the cost of significantly lower recall. This trade-off arises from its strong dependence on correctly identifying intervention targets.

*Table 1.* Experimental results in the "Sachs" dataset.

| real data | Algorithms | pre-edge | rec-edge | F1-edge | pre-orie | rec-orie | F1-orie |
|---|---|---|---|---|---|---|---|
| Sachs | FedCDnv | **0.6128 ± 0.0392** | **0.8096 ± 0.1322** | **0.6967 ± 0.0316** | 0.5045 ± 0.0431 | 0.1846 ± 0.0713 | **0.2861 ± 0.0840** |
| | FedCDnv-vote | 0.4643 ± 0.0124 | 0.7647 ± 0.1369 | 0.5778 ± 0.0998 | 0.3851 ± 0.0959 | 0.2260 ± 0.0124 | 0.2544 ± 0.0384 |
| | CDUIOV | 0.6000 ± 0.0132 | 0.3529 ± 0.0143 | 0.4444 ± 0.143544 | **0.8843 ± 0.0507** | 0.0858 ± 0.0263 | 0.1211 ± 0.0021 |
| | FCI-base | 0.4643 ± 0.0143 | 0.7722 ± 0.0454 | 0.5823 ± 0.1289 | 0.2157 ± 0.0750 | **0.2635 ± 0.0689** | 0.2488 ± 0.0689 |

*Table 2.* Performance of definite causal and non-causal relations.

| $|\mathcal{V}|$ | FDR-C | FDR-dC | FDR-nC | FDR-dnC |
|---|---|---|---|---|
| 10 | **0.0000±0.0000** | **0.0000±0.0000** | 0.2549±0.1061 | **0.2270±0.1169** |
| 20 | **0.0724±0.1766** | 0.0766±0.2458 | **0.0475±0.0291** | 0.0476±0.0247 |
| 30 | 0.0743±0.1313 | **0.0681±0.1703** | 0.0169±0.0181 | **0.0144±0.0168** |
| 40 | 0.1162±0.0898 | **0.0935±0.1176** | 0.0688±0.0098 | **0.0671±0.0093** |
| 50 | 0.1122±0.1073 | **0.0892±0.1458** | 0.0264±0.0051 | **0.0263±0.0043** |
| 60 | **0.0974±0.0964** | 0.1079±0.1180 | 0.0202±0.0045 | **0.0196±0.0047** |
| 80 | **0.0929±0.0809** | 0.1043±0.1101 | 0.0325±0.0026 | **0.0309±0.0029** |
| 100 | 0.1380±0.1008 | **0.1351±0.1655** | 0.0099±0.0020 | **0.0092±0.0018** |
| 120 | 0.0871±0.0619 | **0.0791±0.0966** | 0.0132±0.0019 | **0.0130±0.0019** |
| 150 | 0.0869±0.0485 | **0.0737±0.0546** | 0.0133±0.0012 | **0.0132±0.0013** |

## 4.3. Performance of the Identified Definite Relations

To evaluate the effectiveness of the proposed theories in identifying definite relationships, we compare the FDR values between the returned causal and non-causal relationships and the returned definite causal and definite non-causal relationships. The experimental results are shown in Table 2, where $|\mathcal{V}|$ denotes the number of variables in the underlying DAG. FDR-C and FDR-nC denote the FDR values of the returned causal and non-causal relationships, respectively, while FDR-dC and FDR-dnC represent the FDR values of the returned definite causal and definite non-causal relationships. A lower FDR value indicates better performance.

As shown in Table 2, compared to the identified causal and non-causal relationships, the performances of the extracted definite causal and non-causal relationships are superior in most networks. Specifically, in 6 out of 9 networks, the FDR-dC value is lower than FDR-C, with an average reduction of 1%. In the remaining 3 networks, although the FDR-dC value is slightly higher, the increase is limited to just 0.5%. These results indicate that, on average, definite causal relationships achieve lower FDR values, demonstrating better reliability. For definite non-causal relationships, FedCDnv shows improved performance in 90% of the networks, further confirming the effectiveness of the proposed theories. The consistently lower FDR values for definite relationships, whether causal or non-causal, empirically validate the effectiveness of extracting the "*correct*" relationships.

## 5. Conclusion and Future Work

We propose a novel algorithm, FedCDnv, for discovering causal relationships over non-identical variable sets in federated settings. In this work, we focus on two key aspects. First, we develop theories to examine spurious dependencies caused by the potential presence of non-overlapping variable pairs, extracting definite (or "*correct*") causal and non-causal relationships. Second, we consider the varying importance of causal relationships across different clients and define stable relationships as those that are not only "*correct*" but also "*good*" within each client. To this end, we design a federated strategy—two-level priority selection strategy (*TPSS*)—to aggregate both "*correct*" and "*good*" causal relationships from individual updates, obtaining the global causal graph over an integrated set of variables. Compared to existing baselines, FedCDnv's superior performance demonstrates its effectiveness in federated settings with non-identical variable sets.

In future work, we plan to explore three aspects. First, we aim to develop more effective methods for learning the causal graph from each clients dataset, which contains latent variables and limited sample sizes. Second, recognizing that the current theories for identifying definite relationships are not yet comprehensive, we intend to conduct a more thorough theoretical analysis on each pair of variables. Third, we will also explore additional privacy protection efforts.

## Impact Statement

This paper presents work whose goal is to discover causal relationships from distributed data stored across individual clients (e.g., hospitals), while preserving data privacy. It introduces a more realistic federated setting in which clients observe non-identical variable sets and contribute causal relationships with varying importance, thereby improving the practicality and reliability of causal discovery in real-world applications. While the method avoids direct data sharing, it involves the exchange of structural and statistical information, which may still pose risks of privacy leakage.

## Acknowledgments

This work was supported by the National Natural Science Foundation of China (U24A20323, 62376145, 62376087), the Science and Technology Innovation Talent Team of Shanxi Province (202204051002016), and the Key Technologies Program of Taihang Laboratory in Shanxi Province (THYF-JSZX-24010700).

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

# A. Related Work

In this section, we review and discuss related work from three perspectives. The first perspective involves the discussion of the existing federated causal discovery (FCD) methods. The second perspective reviews the causal discovery (CD) methods from distributed data with non-identical variable sets. And the third perspective presents a detailed discussion about the differences between the above two, that is FCD from decentralized data and CD in distributed setting.

## A.1. Federated Causal Discovery from Decentralized Data

In recent years, to preserve data privacy, some FCD methods have been proposed to discover causal relationships among a set of observed variables from decentralized data. They are categorized into three classes: constraint-based, score-based and continuous-optimization-based methods. Specifically, constraint-based methods utilize conditional independence tests to discover causal relationships among variables in a context of federated learning, such as FedPC (Huang et al., 2023), FedCSL (Guo et al., 2024), and FedCDH (Li et al., 2024b). FedPC introduces a layer-wise aggregation strategy to adapt PC into federated settings. FedCSL designs a federated local-to-global learning strategy, enabling it to scale to high-dimensional data. FedCDH is proposed to accommodate arbitrary causal models and heterogeneous data. For score-based methods, few have been published. DARLS (Ye et al., 2024) utilizes the distributed annealing strategy to search for the optimal graph, while PERI (Mian et al., 2023) aggregates the results of the local greedy equivalent search (GES) (Chickering, 2002) and chooses the worst-case regret for each iteration. Finally, continuous-optimization-based methods mainly extend NOTEARS (Zheng et al., 2018; Deng et al., 2023) with federated strategies. NOTEARS-ADMM (Ng & Zhang, 2022) applies the ADMM (Boyd et al., 2011) optimization to a federated setting; FED-CD (Abyaneh et al., 2022) utilizes belief aggregation to address the problem of inferring causal structure from the mixed data of observational and interventional data; and FedDAG (Gao et al., 2023) learns an adjacency matrix to estimate the DAG and neural representation to approximate causal mechanisms.

All these methods focus on discovering causal relationships over a set of identical variables and do not consider the presence of latent variables, which do not often fit real-world situations where variables measured in different clients are non-identical. In addition, most of them consider the overall quality of the learned causal graph, ignoring the varying importance of causal relationships between variables within this graph.

## A.2. Causal Discovery from Distributed Data

Several methods have been proposed to discover causal relationships from distributed data with non-identical variable sets. Tillman et al. (Tillman et al., 2008) propose the ION algorithm, which discovers a minimal equivalence class of causal DAGs using local independence information on different subsets of variables. To mitigate contradictories resulting from statistical errors, Tillman et al. (Tillman & Spirtes, 2011) propose the IOD algorithm to learn equivalence classes directly by employing statistical tests across datasets simultaneously (Tillman, 2009). Triantafillou et al. (Triantafillou & Tsamardinos, 2015) propose COmbINE, which accepts a collection of interventional datasets over overlapping variable sets under different experimental conditions, and then outputs a summary of all causal models indicating the invariant and variant structural characteristics of all models that simultaneously fit all of the input data sets. To reduce the number of possible structures, Dhir et al. (Dhir & Lee, 2020) adapt and extend bivariate causal discovery algorithms to learn consistent causal structures from multiple datasets with non-identical variable sets. Huang et al. (Huang et al., 2020) propose CD-MiNi to identify causal relationships from multiple observations with non-identical variable sets, under the linearity and non-Gaussianity assumptions. Recently, Cao et al. (Cao et al., 2024) propose the CDUIOV algorithm, designed to discover causal relationships over overlapping variable sets from multiple interventional datasets with unknown intervention targets.

In general, the above algorithms allow the raw data to be shared and ignore the spurious dependencies caused by non-overlapping variables, resulting in the reliability of returned outputs not being guaranteed.

## A.3. Differences between FCD from Decentralized Data and CD from Distributed Data

The differences between federated causal discovery (FCD) from decentralized data and causal discovery (CD) from distributed data mainly stem from data characteristics and application areas.

In a federated setting, data is scattered across multiple independent data sources or clients, and data is not shared between them. Therefore, federated causal discovery focuses on discovering causal relationships between variables from decentralized independent data sources, emphasizing protecting the data privacy and reducing communication costs. It is suitable for

scenarios that require high privacy protection, such as medical and health data (Chengbo & Kai, 2024). In contrast, although distributed data are stored across multiple nodes, these nodes are usually part of a centralized system, and data can flow and be shared between them. Therefore, causal discovery methods for distributed data focus more on the consistency and efficiency of the causal discovery framework. It is suitable for scenarios that require rapid processing and analysis of large amounts of data, such as e-commerce data.

## B. Formal Preliminaries

### B.1. Definitions and Key Concepts

When the causal sufficiency assumption (Spirtes et al., 2000) is violated, the underlying system is typically modeled using SMCMs or MAGs, rather than DAGs. Here, we first introduce the term mixed causal graphs to represent both SMCMs and MAGs. In a mixed causal graph $\mathcal{G} = (\mathcal{V}, \mathcal{E})$, where $\mathcal{V} = \{O, \mathcal{L}, \mathcal{S}\}$, there are directed ($\rightarrow$), bi-directed ($\leftrightarrow$), and undirected ($-$) edges. A directed edge $X \rightarrow Y$ denotes that $X$ is a cause of $Y$, a bi-directed edge $X \leftrightarrow Y$ indicates that there are unobserved confounders between $X$ and $Y$, and an undirect edge $X-Y$ denotes the presence of selection biases between $X$ and $Y$. In $\mathcal{G}$, $X$ is called an ancestor of $Y$ and $Y$ is called a descendant of $X$ if either $X = Y$ or there exists a directed path from $X$ to $Y$. Let $An_{\mathcal{G}}(Y)$ denote the set of ancestors of $Y$ in $\mathcal{G}$. A directed cycle in $\mathcal{G}$ occurs when $X \rightarrow Y \in \mathcal{E}$ and $Y \in An_{\mathcal{G}}(X)$. An almost directed cycle in $\mathcal{G}$ occurs when $X \leftrightarrow Y \in \mathcal{E}$ and either $Y \in An_{\mathcal{G}}(X)$ or $X \in An_{\mathcal{G}}(Y)$. Directed cycles and almost directed cycles are allowed in mixed causal graphs.

We use the MAG $\mathcal{M} = (O, E)$ to model the conditional independence relations among $O$ and use $X_i \perp\!\!\!\perp_{\mathcal{M}} X_j | \mathbf{Z}$ and $X_i \not\perp\!\!\!\perp_{\mathcal{M}} X_j | \mathbf{Z}$ to represent that $X_i$ and $X_j$ are m-separated and m-connected (or conditionally independent and dependent) given a set $\mathbf{Z}$, respectively. Here, m-separation is a natural extension of d-separation, which is presented as Definition B.1. Under the Causal Markov and Faithfulness assumptions (Spirtes et al., 2000), every m-separation present in $\mathcal{M}$ corresponds to a conditional independence and vice-versa.

**Definition B.1. m-separation** (Zhang, 2008; Triantafillou & Tsamardinos, 2015). In $\mathcal{M}$, $X_i$ and $X_j$ are m-separated by $\mathbf{Z} \subseteq O \backslash \{X_i, X_j\}$, if every path between $X_i$ and $X_j$ is blocked by $\mathbf{Z}$. A path $\tau$ between $X_i$ and $X_j$ is blocked by $\mathbf{Z}$ if and only if the following conditions hold: 1) every non-collider on $\tau$ is not a member of $\mathbf{Z}$. 2) every collider on $\tau$ has a descendant in $\mathbf{Z}$.

In general, some orientations of MAGs are indistinguishable from observational data when using constraint-based methods. MAGs with the same set of conditional independencies are considered Markov equivalent, and the complete set of Markov equivalent MAGs is represented by a Partial Ancestral Graph (PAG). In a PAG, endpoints that can be either arrowheads ($>$) or tails ($-$) in different MAGs are denoted with a circle ($\circ$), and the symbol $\star$ serves as a wildcard to denote any of the three marks ($>$, $-$, or $\circ$). There are four types of edges in a PAG: $\rightarrow$, $\circ\!\!\rightarrow$, $\circ\!\!-\!\!\circ$ and $\leftrightarrow$.

### B.2. Connections and Differences among DAGs, SMCMs, MAGs and PAGs

In this subsection, we describe the connections and differences among DAGs, SMCMs, MAGs, and PAGs, which is instantiated as shown in Figure 7. Before that, we first introduce inducing paths, as shown in Definition B.2, which play a pivotal role because adjacencies and non-adjacencies in MAGs can be interpreted as the existence or absence of inducing paths in SMCMs.

**Definition B.2. Inducing path** (Zhang, 2008; Triantafillou & Tsamardinos, 2015). In a mixed causal graph over $\mathcal{V} = O \cup \mathcal{L} \cup \mathcal{S}$, let $X, Y$ ($X, Y \in O$) be any two vertices, and $\mathcal{L}, \mathcal{S}$ be two disjoint sets of latent confounders and selection bias, respectively. A path $p$ between $X$ and $Y$ is called an inducing path relative to $\langle \mathcal{L}, \mathcal{S} \rangle$ if every non-endpoint vertex on $p$ is either in $\mathcal{L}$ or a collider, and every collider on $p$ is an ancestor of either $X$ or $Y$, or a member of $\mathcal{S}$. When $\mathcal{L} = \mathcal{S} = \emptyset$, $p$ is called a primitive inducing path between $X$ and $Y$.

As shown in Figure 7, DAGs, SMCMs, MAGs, and PAGs are different types of causal graphs for describing a system. Specifically, (a) represents a DAG over $O \cup \mathcal{L} \cup \mathcal{S}$, where $O = \{A, B, C, E\}$, $\mathcal{L} = \{L\}$, and $\mathcal{S} = \emptyset$. (b) and (c) are both mixed graphs. (b) represents a SMCM over $O$, which is obtained by hiding the variable $L$ in (a), and (c) represents a MAG over $O$, which is transformed by looking for inducing paths of every pair of variables in (b). (d) is a PAG over $O$, which represents the Markov equivalent class of MAGs with the exact set of conditional independencies.

In Figure 7(b), $A$ and $E$ are not adjacent but they are not m-separated by any set, that is, $A \not\perp\!\!\!\perp E | \emptyset$, $A \not\perp\!\!\!\perp E | \{B\}$, $A \not\perp\!\!\!\perp E | \{C\}$ and $A \not\perp\!\!\!\perp E | \{B, C\}$. Hence, the SMCM skeleton in (b) learned from the data is not unique. To overcome this obstacle, the

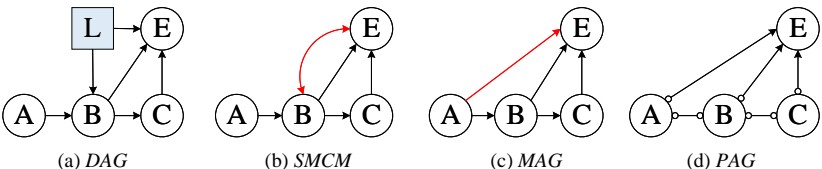

(a) *DAG*         (b) *SMCM*         (c) *MAG*         (d) *PAG*

*Figure 7.* A toy example graph for describing the connections and differences among DAGs, SMCMs, MAGs, and PAGs.

*Table 3.* Summary of notations

| Notations | Meanings |
|---|---|
| $C = \{c_1, \cdots, c_m\}; S$ | $m$ clients; a central server |
| $O; O_k$ | the integrated set of observed variables; the set of variables observed by client $c_k$ |
| $\mathcal{L} \backslash L_k; \mathcal{S}$ | absolute latent variables; relative latent variables $O \backslash O_k$; selection variables |
| $D_k; n_k; d_k$ | data stored in client $c_k$; sample size of $D_k$; number of variables in $O_k$ |
| $\mathcal{M}_k; P_k$ | an underlying MAG describing the causal relationships among $O_k$; the joint probability distribution over $O_k$ |
| $X \perp\!\!\!\perp_{D_k} Y\|\mathbf{Z}; X \not\perp\!\!\!\perp_{D_k} Y\|\mathbf{Z}$ | $X$ and $Y$ are conditionally independent given a set $\mathbf{Z}$ in $D_k$; $X$ and $Y$ are conditionally dependent given a set $\mathbf{Z}$ in $D_k$ |
| $\mathrm{Pag}_0^{c_k}; \mathrm{Pag}_u^{c_k}$ | an initial causal graph learned from the data of client $c_k$; an updated causal graph over $O_k$ |

notation of MAGs is introduced. Here, Figure 7(c) is obtained from Figure 7(b) by looking for primitive inducing paths in (b). Specifically, the path $A \rightarrow B \leftrightarrow E$ is an inducing path in (b), and thus the necessary edge $A \rightarrow E$ is added to (c). In addition, when an almost directed cycle $B \leftrightarrow E$ & $B \rightarrow E$ occurs in (b), the bi-directed edge $B \leftrightarrow E$ is removed from (c). Thus, converting a MAG into the corresponding SMCM is generally not possible, as we cannot know which edges correspond to direct causation or confounding, and which ones are due to primitive inducing paths. Finally, Figure 7(d) is learned using a causal discovery method (e.g., RFCI (Colombo et al., 2012) or ICD (Rohekar et al., 2021)) with an oracle of conditional independence tests. In (d), there are four types of edges: $\rightarrow, \circ\!\!\rightarrow, \circ\!\!-\!\!\circ, \leftrightarrow$. Here, $A\circ\!\!\rightarrow E \leftarrow\!\!\circ C$ is identified as a V-structure, and $B\circ\!\!\rightarrow E$ can be identified using the rules described in (Zhang, 2008).

### B.3. Summary of Notations

We summarize the important symbols and their meanings throughout the paper, as shown in Table 3.

## C. Proofs of the Proposed Theories

Assume that $G_k = (O_k, E_k)$ is a local causal graph learned from the data of client $c_k$ ($k \in \{1, \cdots, m\}$) with an oracle of conditional independence tests. Assume that $G = (O, E)$ is an integrated graph formed by aggregating adjacencies and arrowheads (one of the orientations) from all local graphs (i.e., $O = \bigcup_{k=1}^m O_k$, $E = \bigcup_{k=1}^m E_k$), and $\mathcal{M}$ is the underlying MAG, representing the ground truth graph.

**Theorem C.1.** *Let $\langle X, Y \rangle$ be a non-overlapping variable pair $(X, Y \in O)$ and $\mathcal{A}_X^G$ be the set of variables adjacent to $X$ in $G$. If there exists $\mathcal{A}_X^G \cap \mathcal{A}_Y^G = \emptyset$ & $\mathcal{A}_X^G \neq \emptyset$ & $\mathcal{A}_Y^G \neq \emptyset$, and for each $Y' \in \mathcal{A}_Y^G$, $\langle X, Y' \rangle$ is not a non-overlapping variable pair (and vice versa), then the relationship between $X$ and $Y$ is considered definitively non-causal.*

*Proof.* Since $X$ and $Y$ are not observed simultaneously by any client, their relationship cannot be identified from any dataset. In other words, whether $X$ and $Y$ are adjacent in the ground truth MAG $\mathcal{M}$ remains unknown. In addition, we assume that for each $Y' \in \mathcal{A}_Y^G$, $\langle X, Y' \rangle$ is not a non-overlapping variable pair (and vice versa), and consider two possible relationships between $X$ and $Y$ in $\mathcal{M}$, as follows.

(1) Assume that $X$ and $Y$ are not adjacent in $\mathcal{M}$. The condition "$\mathcal{A}_X^G \cap \mathcal{A}_Y^G = \emptyset$ & $\mathcal{A}_X^G \neq \emptyset$ & $\mathcal{A}_Y^G \neq \emptyset$" implies that either there are no paths between $X$ and $Y$, or there exists paths that involve at least two intermediary nodes, apart from $X$ and $Y$. Therefore, the proposed conditions are met in this case, and we conclude that the relationship between $X$ and $Y$ is non-causal.

(2) Assume that $X$ and $Y$ are adjacent in $\mathcal{M}$. The condition "$\mathcal{A}_X^G \cap \mathcal{A}_Y^G = \emptyset$" implies three possible cases.

    (a) In the ground truth graph $\mathcal{M}$, $X$ is adjacent only to $Y$, and vice versa. That is, $X$ and $Y$ are mutually adjacent in $\mathcal{M}$. Since $\langle X, Y \rangle$ is a *non-overlapping variable pair*, we learn $\mathcal{A}_X^G = \emptyset$ and $\mathcal{A}_Y^G = \emptyset$ from the data, leading to $\mathcal{A}_X^G \cap \mathcal{A}_Y^G = \emptyset$.

(b) $\exists A \in O \backslash \{X, Y\}$ such that $X \circ \rightarrow Y \leftarrow \circ A$ (or $Y \circ \rightarrow X \leftarrow \circ A$). Since $X$ and $Y$ are never observed together by any client, $\mathcal{A}_X^G = \emptyset$ (or $\{A\}$) and $\mathcal{A}_Y^G = \{A\}$ (or $\emptyset$), which leads to $\mathcal{A}_X^G \cap \mathcal{A}_Y^G = \emptyset$.

(c) $\exists A, B \in O \backslash \{X, Y\}$ such that $A \circ \rightarrow X \leftarrow \circ Y \circ \rightarrow B$. In this case, $\mathcal{A}_X^G = \{A, B\}$ and $\mathcal{A}_Y^G = \{B\}$. However, if $X$ and $B$ form a *non-overlapping variable pair*, then $\mathcal{A}_X^G = \{A\}$ and $\mathcal{A}_Y^G = \{B\}$, and their intersection is empty.

To avoid the cases (a) and (b), the condition that "$\mathcal{A}_X^G \neq \emptyset$ & $\mathcal{A}_Y^G \neq \emptyset$" is required. For case (c), the condition that "for each $Y' \in \mathcal{A}_Y^G$, $\langle X, Y' \rangle$ is not a *non-overlapping variable pair* (vice versa)" is presented.

Based on the above analysis, it is sufficient to prove that $X$ and $Y$ are not adjacent in the underlying graph $\mathcal{M}$ using the mentioned conditions, thereby inferring the non-causal relationship between $X$ and $Y$. □

**Lemma C.2.** *Assume that $X$ and $Y$ ($X, Y \in O$) are observed simultaneously by at least one client and $\mathbf{Z}_n$ represents a union set of variables that appear in non-overlapping variable pairs with non-definite, non-causal relationships. If $X \in \mathbf{Z}_n$ or $Y \in \mathbf{Z}_n$, then the causal relationship between $X$ and $Y$ is considered to be non-definite.*

*Proof.* $X$ and $Y$ are observed simultaneously by at least one client, implying that $\langle X, Y \rangle$ is not a *non-overlapping variable pair*. Suppose there exists a variable $A \in O \setminus \{X, Y\}$ such that $X$ (or $Y$) and $A$ form a *non-overlapping variable pair* and their relationship is considered to be non-definite and non-causal, which implies that $X \in \mathbf{Z}_n$ or $Y \in \mathbf{Z}_n$. There are two cases regarding the relationship between $X$ and $Y$.

Case 1: $X$ and $Y$ are adjacent in the ground truth $\mathcal{M}$. The relationship between $X$ and $Y$ is inferred to be adjacent because no set $\mathbf{Z}$ exists such that $X$ and $Y$ are conditionally independent given $\mathbf{Z}$, regardless of whether $X$ and $A$ are adjacent in $\mathcal{M}$.

Case2: $X$ and $Y$ are not adjacent in the ground truth $\mathcal{M}$. Since $\langle X, A \rangle$ (or $\langle Y, A \rangle$) forms a *non-overlapping variable pair* with a non-definite, non-causal relationship, $X$ and $Y$ may be learned as adjacent from local data, which conflicts with the non-adjacency between $X$ and $Y$ in $\mathcal{M}$. This occurs because $A$ could be a confounder for $X$ and $Y$, leading to the learned adjacency between them. As a result, the causal relationship between $X$ and $Y$ is considered to be non-definite.

In summary, we conclude that if $X \in \mathbf{Z}_n$ or $Y \in \mathbf{Z}_n$, then the causal relationship between $X$ and $Y$ is considered to be non-definite. □

**Lemma C.3.** *Assume that $X$ and $Y$ ($X, Y \in O$) are observed simultaneously by at least one client and $\mathcal{A}_X^G$ is a set of variables adjacent to $X$ in $G$. If there exists a non-overlapping variable pair $\langle A, B \rangle$ such that $\{A, B\} \subseteq \mathcal{A}_X^G \cup \mathcal{A}_Y^G$, then the causal relationship between $X$ and $Y$ is considered to be non-definite.*

*Proof.* Since $X$ and $Y$ are observed together by at least one client, their relationship can be learned from at least one local dataset. Assume that $X$ and $Y$ are not adjacent in the ground truth graph $\mathcal{M}$, and then $\{A, B\} \subseteq \mathcal{A}_X^G \cup \mathcal{A}_Y^G$ implies three possible cases regarding the structures of $\langle A, B \rangle$ and $\langle X, Y \rangle$ in $\mathcal{M}$.

Case1: Both $A$ and $B$ are linked to both $X$ and $Y$ in $\mathcal{M}$. In this case, the separating set that makes $X$ and $Y$ conditionally independent must be a subset of the adjacent set of either $X$ or $Y$. If there exists a set $\mathbf{Z}$ such that $X$ and $Y$ independent given $\mathbf{Z}$, and $\mathbf{Z}$ contains $\{A, B\}$, then based on the dataset $D_k$ ($k \in 1, \cdots, m$), we would learn that $X$ and $Y$ are not m-separated by $\mathbf{Z}$. The reason is that $\mathbf{Z} \subseteq \mathcal{A}_X^G \cup \mathcal{A}_Y^G$, $\{A, B\} \subseteq \mathbf{Z}$, but $\{A, B\} \nsubseteq O_k$ ($k \in 1, \cdots, m$). Thus in this case, $X$ and $Y$ are adjacent in the learned causal graph $G_k$, which conflicts with the fact that $X$ and $Y$ are not adjacent in $\mathcal{M}$.

Case2: Either $A$ or $B$ is linked to both $X$ and $Y$ in $\mathcal{M}$. Suppose that $A$ is adjacent to $X$, and $B$ is adjacent to $Y$. Since the separating sets for $X$ and $Y$ are incomplete, we may learn that $X$ and $Y$ are adjacent in $G_k$, which would be inconsistent with their non-adjacency in $\mathcal{M}$.

Case3: Neither $A$ nor $B$ is linked to $X$ or $Y$. In this case, if $\{A, B\} \nsubseteq \mathcal{A}_X^G \cup \mathcal{A}_Y^G$, then the separating sets for $X$ and $Y$ are complete. As a result, the learned causal relationship between $X$ and $Y$ is consistent with the ground truth graph $\mathcal{M}$.

In summary, when there exists a *non-overlapping variable pair* $\{A, B\} \subseteq \mathcal{A}_X^G \cup \mathcal{A}_Y^G$, the relationship between $X$ and $Y$ learned from the data may differ from those in the ground truth and is considered to be non-definite. □

# D. Experiments

This section provides a detailed overview of the experiments. Section D.1 describes the problem attributes and presents the details of the baseline methods. Section D.2 displays additional experimental results on precision and recall, comparing

*Table 4.* Test values and default values of problem attributes in generating experiments in each iteration of FedCDnv.

| Problem attributes | Test values | Default value |
|---|---|---|
| Number of variables generated in the underlying DAG ($|\mathcal{V}|$) | {10, 20, 30, 40, 50, 60, 80, 100} | 30 |
| Ratio of the number of latent variables to variables in underlying DAGs ($\lambda = \frac{|\mathcal{V}| - |\mathcal{O}|}{|\mathcal{V}|}$) | {5%, 10%, 15%, 20%, 25%, 30%, 35%} | 15% |
| Number of clients ($m$) | {2, 4, 6, 8, 10, 15, 20, 30} | 6 |
| Ratio of the number of variables per dataset to the integrated variables ($\delta = \frac{|\mathcal{O}_i|}{|\mathcal{O}|}$) | {95%, 90%, 85%, 80%, 75%, 70%, 65%, 60%} | 85% |
| Interval of confidence level $[\alpha - \theta_1, \alpha + \theta_2]$ with varying $\alpha - \theta_1$, where $\alpha = 0.05$ and $\alpha + \theta_2 = 0.5$ | {0.001, 0.002, 0.003, 0.004, 0.005, 0.006, 0.007, 0.008, 0.009} | [0.001, 0.5] |
| Interval of confidence level $[\alpha - \theta_1, \alpha + \theta_2]$ with varying $\alpha + \theta_2$, where $\alpha = 0.05$ and $\alpha - \theta_1 = 0.001$ | {0.5, 0.45, 0.4, 0.35, 0.3, 0.25, 0.2, 0.15, 0.1} | [0.001, 0.5] |
| Interval of Sample size per dataset ($[100, ns]$) | {200, 500, 1000, 2000, 3000, 5000} | [100,2000] |

*Table 5.* Description of benchmark BNs.

| Network | Num.Nodes | Num.Edges | Max In/Out Degree | Min/Max |PC| | Type |
|---|---|---|---|---|---|
| Child | 20 | 25 | 2/7 | 1/8 | Medium |
| Insurance | 27 | 52 | 3/7 | 1/9 | Medium |
| Alarm | 37 | 46 | 4/5 | 1/6 | Medium |
| Barley | 48 | 84 | 4/5 | 1/8 | Medium |
| Child3 | 60 | 79 | 3/7 | 1/8 | Large |
| Alarm3 | 111 | 149 | 4/5 | 1/6 | Very-Large |

FedCDnv, FedCDnv-vote, CDUIOV, CD-MiNi, and FCI-base. Section D.3 presents and analyzes the experimental results, comparing our method with existing FCD methods, including FedACD, FedPC, and Notears-ADMM. Section D.4 assesses the sensitivity of stable relationships to the performance of FedCDnv. Finally, Section D.5 presents the impact of different parameters on the performance of FedCDnv.

### D.1. More Details of Experimental Settings

First, for each invocation of the algorithm, the problem instance (i.e., the synthetic data) is generated using the parameters shown in Table 4. Then, Figure 5 shows the details of the benchmark networks used in the experiments. Finally, to comprehensively verify the effectiveness of the proposed method, we conduct two types of experiments. The descriptions of the all compared methods (including ours) are presented as follows.

For the first type of experiments, the compared methods include FedCDnv, FedCDnv-vote, CDUIOV (Cao et al., 2024), CD-MiNi (Huang et al., 2020), and FCI-base. FedCDnv is our proposed method. FedCDnv-vote is a variant of FedCDnv that incorporates a voting strategy. CDUIOV is a constraint-based method that discovers causal structures from interventional datasets with overlapping variable sets. CD-MiNi is a well-known SCM-based method that identifies causal relationships over the integrated set of variables in linear, non-Gaussian settings. FCI-base is a baseline method that integrates multiple causal structures learned by FCI using a sample-size-based federated strategy in the context of federated learning.

For the second type of experiments, the compared methods include FedCDnv, FedCDnv-vote, Notears-ADMM (Ng & Zhang, 2022), FedPC (Huang et al., 2023), and FedCSL (Guo et al., 2024). FedCDnv and FedCDnv-vote are our proposed methods, which differ in their federated strategies. Notears-ADMM, FedPC and FedCSL are based on the assumptions that all clients observe identical variable sets and that there are no latent variables—assumption that differ from ours. Thus, the comparison is intended only to provide a rough performance reference rather than a strict benchmarking. Consequently, we evaluate only the performance on the learned skeletons. Specifically, Notears-ADMM is a continuous-optimization-based method in the federated learning setting. FedPC and FedCSL are two constraint-based federated CSL methods. FedPC presents a layer-wise aggregation strategy for a seamless adaptation of the PC algorithm into the federated learning paradigm. FedCSL first infers the relative sample sizes held by each client, and then performs a weighted aggregation of the learned structures from each client using weights based on their sample sizes.

### D.2. Additional Details on the Comparison of FedCDnv, FedCDnv-vote, CDUIOV, CD-MiNi, and FCI-base

Figures 8, 9 present detailed experimental results for precision and recall of learned edges and orientations, comparing FedCDnv, FedCDnv-vote, CDUIOV, CD-MiNi, and FCI-base.

Figure 8 presents the experimental results of FedCDnv and CD-MiNi on networks with 5–10 nodes. It is observed that

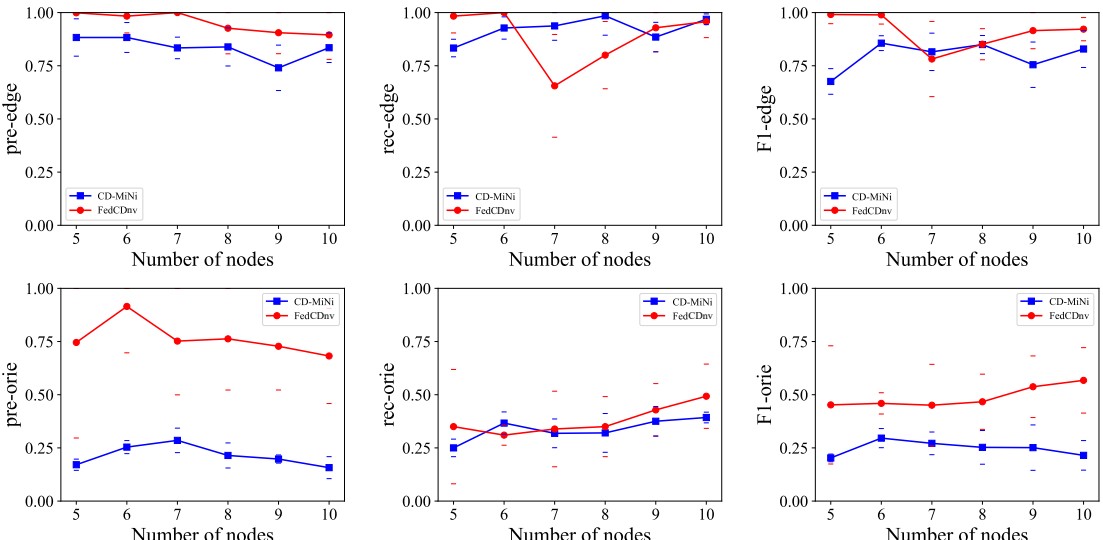

*Figure 8.* The setting of non-identical variable sets: FedCDnv vs. CD-MiNi.

compared with CD-MiNi, FedCDnv significantly outperforms CD-MiNi in terms of precision for learned edges (pre-edge) and orientations (pre-orie), with the pre-orie obtained by FedCDnv being nearly 30% better than CD-MiNi. However, the recall values of edges and orientations are almost comparable between FedCDnv and CD-MiNi. For the recall values of returned edges (rec-edge), FedCDnv outperforms CD-MiNi on some networks, while CD-MiNi performs better on others. As for the recall value of learned orientations (rec-orie), both methods are comparable, with FedCDnv slightly outperforming CD-MiNi. Therefore, considering the F1 scores for edges (F1-edge) and orientations (F1-orie), FedCDnv demonstrates superior performance. The reasons for the difference between the two methods are discussed in Section 4.2.

Figure 9 displays the detailed experimental results among FedCDnv, FedCDnv-vote, CDUIOV, and FCI-base. It is observed that FedCDnv shows a significant performance advantage in learning orientations, with pre-orie and rec-orie values noticeably higher than those of the competing algorithms. In edge learning, FedCDnv outperforms the competing algorithms slightly. Specifically, the first two rows of Figure 9 show experimental results of five algorithms on networks with different scales. The results show that, for pre-edge, FedCDnv slightly outperforms other algorithms, while for rec-edge, the five algorithms are comparable with little difference. As for the returned orientations, the pre-orie and rec-orie obtained by FedCDnv significantly outperform the competing algorithms in most networks. Next, we analyze the impact of varying numbers of clients on the algorithm's performance, shown in the third and fourth rows of Figure 9. Similar to the results on networks of different scale (i.e., the first two rows of Figure 9), FedCDnv clearly outperforms the competing algorithms in learning orientations. Furthermore, FedCDnv shows more notable superiority in precision of learned edges (pre-edge), outperforming the competing algorithms. Notably, CDUIOV shows more significant performance variation in pre-edge across different network scales and client numbers compared with the other algorithms. This may be due to the fact that CDUIOV requires intervention data, and the influence of different data sources is more pronounced. Finally, we analyze the experimental results of the five algorithms on the benchmark data, as shown in the last two rows of Figure 9. We observe that the advantages of FedCDnv in terms of pre-edge and rec-edge are not very pronounced, but its performance in pre-orie and rec-orie is better overall, clearly outperforming other algorithms. Overall, the superior experimental results of FedCDnv demonstrate its effectiveness in causal structure learning with non-identical variable sets, when compared with other distributed CSL methods in the same setting.

Moreover, based on Figure 9, we also analyze the effect of different network scales and numbers of clients on FedCDnv. For the former, shown in the first two rows of Figure 9, we observe a general decreasing trend in the performance of FedCDnv as the number of nodes increases, except for pre-edge. For networks with 30 or more nodes, the value of pre-edge remains almost constant, while the value of rec-edge decreases as the number of nodes increases. Although the rec-edge obtained by FedCDnv reaches its lowest value for the network with 50 nodes, this may be due to errors in CI testing. As a result, the F1-edge of FedCDnv decreases as the increasing of the network scale. Regarding the learned orientations, the value of pre-orie decreases significantly as the number of nodes increases, whereas rec-orie remains relatively stable, except

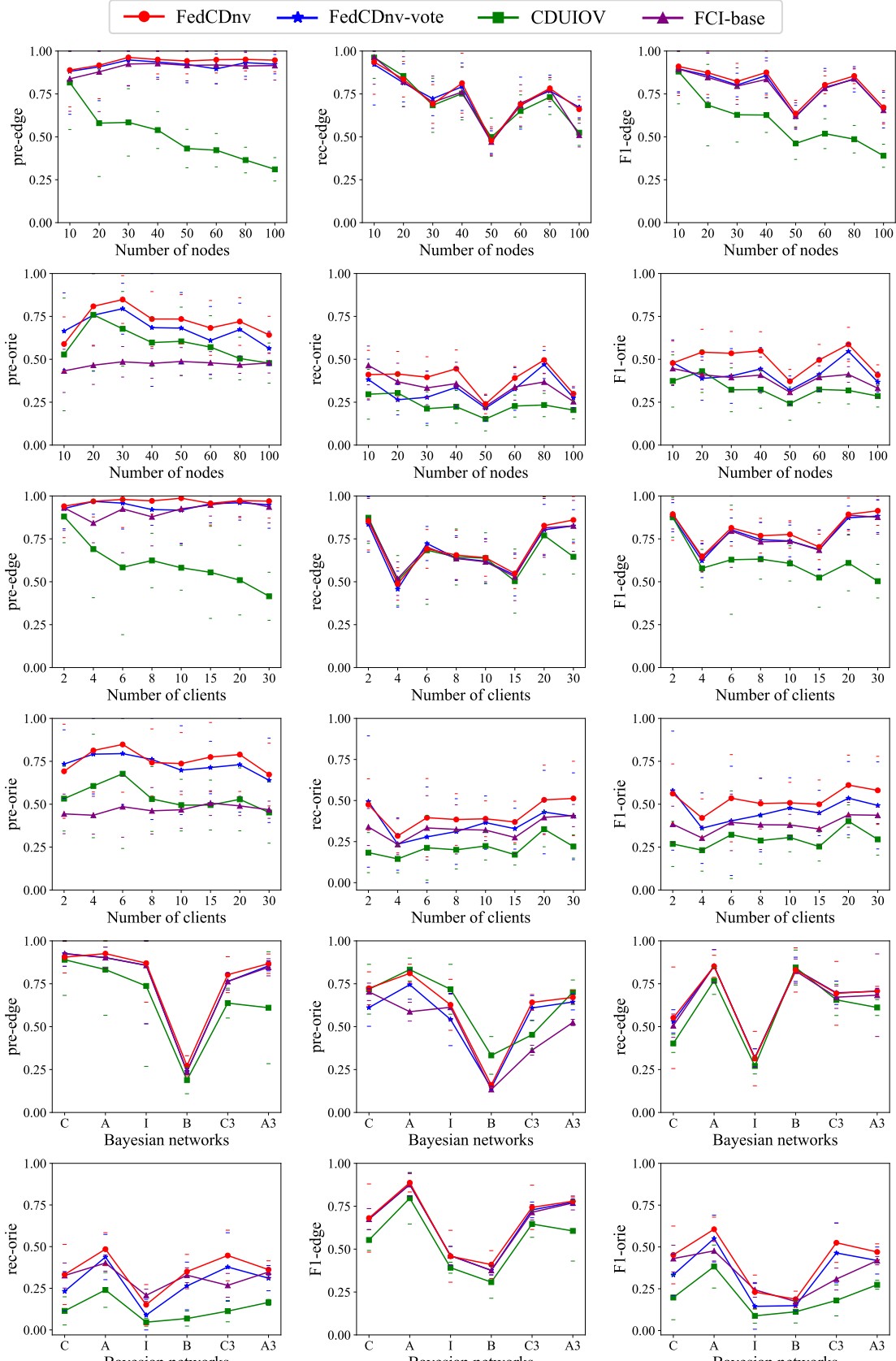

*Figure 9.* The setting of non-identical variable sets: FedCDnv vs. FedCDnv-vote vs. CDUIOV vs. FCI-base.

for networks with 50 nodes. Therefore, the overall F1-orie also exhibits a decreasing trend. Regarding the impact of the number of clients on FedCDnv, we refer to the results shown in the panels of the third and fourth rows of Figure 9. It can be observed that as the number of clients increases, the overall performance of FedCDnv in terms of both edges and orientations improves. Specifically, regarding edge learning performance, pre-edge shows minimal variation across different numbers of clients, while the value of rec-edge increases significantly. Consequently, the overall metric F1-edge also increases as the number of clients grows. For learned orientations, apart from the case where the number of clients is 2—in which case FedCDnv achieves relatively high values for pre-orie, rec-orie, and F1-orie—the values of these three metrics generally increase as the number of clients increases. This improvement in the metrics evaluating learned orientations is more gradual compared to the improvement in edge learning performance.

### D.3. Compare FedCDnv, FedCDnv-vote, Notears-ADMM, FedCSL, and FedPC

In this section, we compare the experimental performance of our method with that of existing FCD algorithms on the learned causal skeleton, assuming that multiple clients observe the entirely identical variables.

*Synthetic Data*. The first row of Figure 10 presents the experimental results of FedCDnv and other FCD methods on synthetic data with node counts ranging from 10 to 100. It is observed that, first, compared with FedCDnv-vote, the pre-edge value of FedCDnv is significantly higher than that of FedCDnv-vote in the presented networks, while the rec-edge values for the two methods are comparable. According to the F1-edge metric, FedCDnv demonstrates better skeleton learning performance in networks with 10, 20, 50, 80, and 100 nodes. In the remaining networks, the performance of both methods is similar. These results indicate the effectiveness of the proposed federated strategy *TPSS*. In addition, compared with the other three FCD algorithms, the pre-edge value returned by FedCDnv is slightly inferior to that of FedCSL but significantly better than FedPC and Notears-ADMM. Moreover, the rec-edge value returned by FedCDnv is significantly higher than the others, outperforming them by nearly 25%. Therefore, according to the F1-edge metric, FedCDnv outperforms significantly Notears-ADMM in networks with fewer variables. As the number of variables increases, the performance gap between the two methods gradually narrows. Nevertheless, in networks with 80 and 100 nodes, FedCDnv still outperforms Notears-ADMM by approximately 5%. In comparison with FedPC and FedCSL, FedCDnv obviously performs better. This can be attributed to the fact that, in this comparison, we assume the presence of *absolute latent variables*. Our method uses the FCI algorithm to learn the local causal graph for each client and aggregates the results in a federated setting, producing a PAG. In contrast, FedPC and FedCSL operate under the causal sufficiency assumption and use the CD algorithm (e.g., PC) to return the Markov equivalence class of DAGs. As a result, the inconsistent assumptions lead to poorer skeleton learning performance in these two algorithms when applied to data with latent variables.

The second row of Figure 10 presents the experimental results of FedCDnv and other FCD algorithms on synthetic data with varying numbers of clients. The results show that FedCDnv consistently returns the highest pre-edge values across different client counts. Furthermore, as the number of clients increases, the pre-edge values of compared methods remain relatively stable, except for Notears-ADMM, which experiences a significant drop when the number of clients reaches 30. Other algorithms' pre-edge values show only a slight decline at 30 clients. As for the rec-edge, it can be seen that FedCDnv and FedCDnv-vote perform similarly, and both outperform FedPC and FedCSL. The reason for this, as mentioned above, is primarily differences in the underlying assumptions of the algorithms. It is also noteworthy that the rec-edge value obtained by Notears-ADMM increases with the number of clients and stabilizes after reaching 8 clients. In summary, from the F1-edge metric, the performance of both FedCDnv and FedCDnv-vote remains stable as the number of clients increases, consistently outperforming the other three algorithms. In contrast, the performance of Notears-ADMM gradually improves with the increasing number of clients. After reaching 8 or more clients, its performance stabilizes, but when the number of clients reaches 30, the F1-edge value begins to decline. This suggests that Notears-ADMM may have an optimal range of client numbers.

*Benchmark Data*. The last row of Figure 10 shows the experimental results of FedCDnv and other FCD methods on six benchmark networks. Specifically, FedCDnv performs similarly to FedCDnv-vote in four networks, while in the other two networks, "Insurance" and "Barley", FedCDnv significantly outperforms FedCDnv-vote. Compared with FedPC and FedCSL, FedCDnv shows a clear advantage in five of the networks, but in the "Barley" network, FedCDnv performs slightly worse than FedCSL. The gap is mainly due to the pre-edge learned by FedCSL, which exceeds 70%, significantly higher than the approximately 50% achieved by other algorithms, making FedCSL the most optimal in the "Barley" network. Finally, it is also observed that Notears-ADMM performs poorly across the benchmark networks. Although its performance in the "Barley" network is close to that of the other algorithms, in the remaining networks, Notears-ADMM is significantly inferior to the other algorithms. This may be due to the unknown data generation mechanism in the benchmarks, which

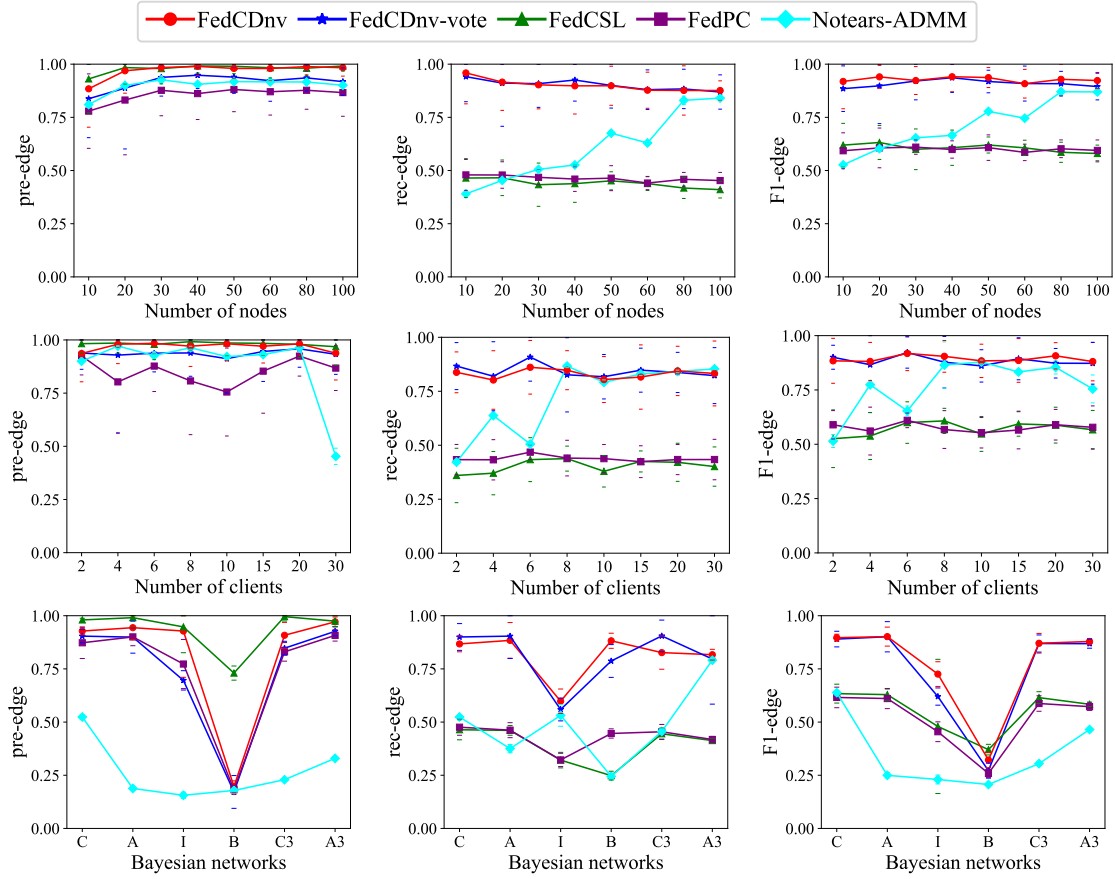

*Figure 10.* The setting of identical variable sets: Experimental results in synthetic and benchmark data.

*Table 6.* Experimental results in the "Sachs" dataset.

| real data | Algorithms | pre-edge | rec-edge | F1-edge | pre-orie | rec-orie | F1-orie |
|---|---|---|---|---|---|---|---|
| Sachs | FedCDnv | $0.7222 \pm 0.0139$ | **$0.6500 \pm 0.1076$** | **$0.68 \pm 0.1199$** | $0.3846 \pm 0.0689$ | $0.2500 \pm 0.0032$ | **$0.3030 \pm 0.0217$** |
| | FedCDnv-vote | $0.5152 \pm 0.0573$ | **$0.6500 \pm 0.0886$** | $0.6115 \pm 0.1231$ | $0.3143 \pm 0.0522$ | $0.2500 \pm 0.0032$ | $0.2633 \pm 0.0332$ |
| | Notears-Admm | **$0.7731 \pm 0.0638$** | $0.4091 \pm 0.0226$ | $0.4304 \pm 0.1435$ | | | |
| | FedCSL | $0.7692 \pm 0.0638$ | $0.2500 \pm 0.0096$ | $0.3774 \pm 0.1131$ | | | |
| | FedPC | $0.5862 \pm 0.0638$ | $0.4250 \pm 0.0133$ | $0.4928 \pm 0.1289$ | | | |

leads to slightly suboptimal performance of Notears-ADMM in certain networks.

***Real-world Data.*** Table 6 presents experimental results of FedCDnv and other FCD methods on the real-world data "Sachs", where the variable sets observed by all clients are assumed to be identical. In addition, since FedCDnv assumes the existence of *absolute latent variables*, we only compare the correctness of the learned skeleton against existing FCD methods. It is observed that, according to the overall F1-edge metric, FedCDnv shows the best performance. However, in terms of edge precision (pre-edge), Notears-ADMM performs the best, followed by FedCSL, and then FedCDnv, which lags behind Notears-ADMM by nearly 5%. Meanwhile, in terms of rec-edge, FedCDnv and FedCDnv-vote outperform Notears-ADMM by nearly 20% and FedCSL by nearly 40%. Overall, FedCDnv demonstrates the superior performance, highlighting the effectiveness of the proposed federated strategy on the "Sachs" dataset.

### D.4. Sensitivity Analysis of Stable Relationships on the Performance of FedCDnv

For the parameter $[\alpha - \theta_1, \alpha + \theta_2]$, we first evaluate the impact of varying $\alpha - \theta_1$ on the performance of FedCDnv, while keeping $\alpha + \theta_2 = 0.5$. The experimental results are shown in the left subplot of Figure 11. It can be observed that as the parameter $\alpha - \theta_1$ increases, the performance of the FedCDnv algorithm exhibits a specific pattern of variation. When $\alpha - \theta_1$ is less than 0.003, the performance improves gradually. However, after $\alpha - \theta_1$ exceeds 0.003, its performance begins to

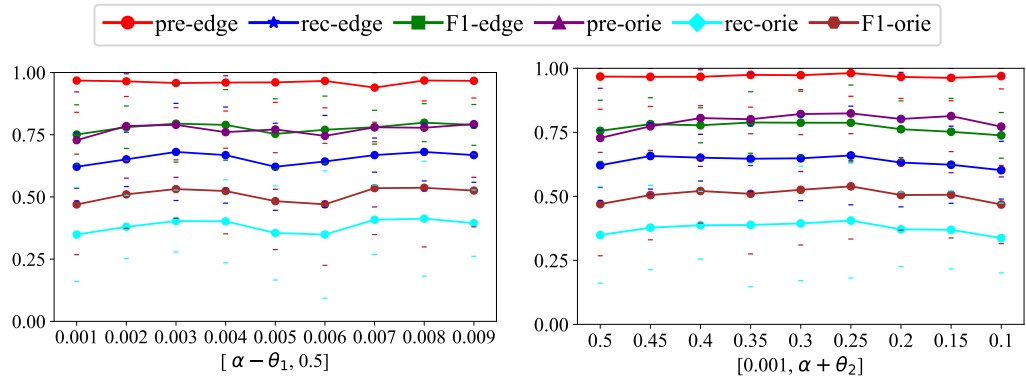

*Figure 11.* The performance of FedCDnv with varying $[\alpha - \theta_1, \alpha + \theta_2]$.

decline, reaching its lowest point at $\alpha - \theta_1 = 0.005$ and 0.006. Subsequently, the performance rises again at $\alpha - \theta_1 = 0.007$ and $\alpha - \theta_1 = 0.008$, but starts to decline once more when $\alpha - \theta_1$ reaches 0.009. In summary, the performance of FedCDnv fluctuates as $\alpha - \theta_1$ changes, following a pattern of "rising, falling, rising again, and finally falling."

Similarly, we evaluate the impact of varying $\alpha + \theta_2$ while keeping $\alpha - \theta_1 = 0.001$, on the performance of FedCDnv. The experimental results are shown in the right subplot of Figure 11. It is observed that as the interval narrows, most metrics—rec-edge, F1-edge, pre-orie, rec-orie, and F1-orie—exhibit a trend resembling a normal distribution: higher values in the middle and lower values on both ends, except for the pre-edge metric, which remains relatively stable. This indicates that an appropriately chosen confidence interval can enhance the performance of FedCDnv, while intervals that are either too small or too large are less effective for establishing stable relationships.

In conclusion, the parameters $\theta_1$ and $\theta_2$ have a moderate impact (5%-10%) on the performance of FedCDnv. It is notably that FedCDnv exhibits higher sensitivity to changes in $\alpha + \theta_2$.

### D.5. Effects of Different Parameters on the Performance of FedCDnv

In this subsection, we evaluate the impact of varying values of the parameters listed in Table 4 (i.e., problem attributes) on the performance of FedCDnv. During the evaluation of each parameter, all other parameters are kept at their default values. The tested parameters include: the number of nodes in the underlying DAG ($|\mathcal{V}|$), the number of clients ($m$), the proportion of each client's observed variables relative to the integrated variables ($\delta$), the proportion of *absolute latent variables* ($\lambda$), the sample size per client ([100, $ns$] ranging from 100 to $ns$), and the interval of confidence level ($[\alpha - \theta_1, \alpha + \theta_2]$ with $\alpha = 0.05$). The experimental results and analysis for $|\mathcal{V}|$ and $m$ are presented in Section D.2, while those for the confidence interval $[\alpha - \theta_1, \alpha + \theta_2]$ are given in Section D.4. Results for the remaining parameters are summarized in Figure 12.

For the parameter $\delta$ (left subfigure of Figure 12), the results show a general decreasing trend in the performance of FedCDnv for both edges and orientations as $\delta$ increases. Specifically, in terms of edge learning, the precision for edges (pre-edge) gradually declines, while the recall for edges (rec-edge) slightly increases with larger $\delta$ values, resulting in a relatively stable F1-edge score. In contrast, for orientation learning, the precision (pre-orie), recall (rec-orie), and F1 score (F1-orie) all show a consistent downward trend as $\delta$ increases. These observations indicate that a lower overlapping rate among observed variables negatively affects the orientation learning capability of FedCDnv.

The impact of the parameter $\lambda$ on FedCDnv is shown in the middle subfigure of Figure 12. The results indicate that as $\lambda$ increases from 0.05 to 0.2, the performance metrics for both edges and orientations exhibit a declining trend. However, when $\lambda$ increases further from 0.2 to 0.35, the performance improves noticeably. Specifically, the precision for edges (pre-edge) remains relatively stable across different $\lambda$ values, while the precision for orientations (pre-orie) gradually decreases as $\lambda$ rises to 0.2. Other metrics—including rec-edge, F1-edge, rec-orie, and F1-orie—drop sharply as $\lambda$ approaches 0.2. Interestingly, we observe that from $\lambda = 0.25$ to 0.35, the performance of FedCDnv in both edge and orientation learning improves significantly. Overall, the performance of FedCDnv tends to degrade as $\lambda$ increases.

Lastly, we investigate the impact of the sample size per client, $n_i \in [100, ns]$ and $ns \in \{200, 500, 1000, 2000, 3000, 5000\}$, on the performance of FedCDnv, focusing on how the evaluated metrics vary with changes in $ns$. The experimental results are presented in the right subfigure of Figure 12. It is observed that as $ns$ increases, the performance of FedCDnv improves

*Figure 12.* Experimental results of FedCDnv with different parameters.

significantly, with rec-edge, F1-edge, rec-orie, and F1-orie all exhibiting upward trends. In contrast, the pre-edge reaches its peak at $ns = 2000$ and then declines slightly as $ns$ continues to increase. Similarly, pre-orie exhibits a trend similar to pre-edge, peaking at $ns = 1000$ and maintaining relative stability as $ns$ increases further.

