# OpenReview forum: "Federated Causal Structure Learning with Non-identical Variable Sets"
_ICML.cc/2025/Conference — ICML 2025 poster_

### Official Review · Reviewer_mhPQ · 2025-03-13

**Overall Recommendation:** 2

**Summary:**

The paper introduces FedCDnv, a novel algorithm for federated causal discovery where clients observe non-identical but overlapping variable sets. A key challenge in this scenario is the spurious dependencies introduced by non-overlapping variables. To address this, the paper proposes a two-level priority selection strategy that aggregates local graphs from each client to form a global causal graph. The paper demonstrates the effectiveness of FedCDnv through extensive experiments on synthetic, benchmark, and real-world datasets, showing improvements over existing methods.

**Claims And Evidence:**

Some claims in the paper are not fully supported by strong evidence. For example, the authors claim that FedCDnv can achieve federated causal discovery while "preserving data privacy" (line 93-94).
This claim seems over-claimed since the paper only mentions that "FedCDnv exchanges structural information rather than raw data, protecting data privacy to a certain extent", without any specific designs for the privacy concerns.

Another claim is that FedCDnv works even "when the sample distributions differ across clients" (Assumption 2.2, lines 101-105). However, the paper does not offer theoretical analysis or empirical results to support this point.

**Essential References Not Discussed:**

N/A

**Experimental Designs Or Analyses:**

In Assumption 2.2, the paper claims that FedCDnv works when the sample distributions differ across clients. However, there is no empirical evidence to support this claim.

In Section 4.3, the authors use the false discovery rate (FDR) as an evaluation metric for definite relationships. It would be helpful to explain why FDR is chosen over other metrics and to also include results on recall for a more complete evaluation.

The reults about the performance across different numbers of clients are presented in appendix. These results could be moved to the main paper, as the number of clients is an important factor in federated leanring and deserves more emphasis.

**Methods And Evaluation Criteria:**

Methods:
The idea of aggregating both "good" and "correct" relationships using the concept of "stable relationships" is novel and well-motivated. However, some details need further explanation:
* In Algorithm 1 (line 7) and lines 250-251, the paper claims to integrate all local PAGs into a global graph by taking the union of nodes and edges. However, there is no analysis of this union operation. Consider the following situations:
  * If a client learns a wrong edge, it becomes part of the global graph. What effect does this have on the final result?
  * When different clients report conflicting directions for an edge, how is the conflict resolved? Is it merged into a bi-directional edge, and how does this impact the outcome?
* In Algorithm 2, lines 9–12, the paper applies an "orientation rule" on the client side. The rationale behind this rule is not sufficiently explained, and further analysis is needed to clarify its effectiveness and correctness.

Evaluation Criteria:
The paper uses False Discovery Rate (FDR) to evaluate the effectiveness of FedCDnv in identifying the definite causal and non-causal relationships in Sec 4.3. It would be helpful to explain why FDR is chosen over other metrics and to also include results on recall for a more complete evaluation.

**Other Comments Or Suggestions:**

N/A

**Other Strengths And Weaknesses:**

There are certain points that need to be clarified in the paper.
* In Section 3.2.1 (lines 179-181), the paper states that: "The first level is to determine whether the adjacency between X_i and X_j is caused by relative latent variables, which is detected by Lemma 3.4". However, Lemma 3.4 only accounts for bidirected edges. What happens if the adjacency between X_i and X_j is not a bidirected edge?
* In Section 3.2.1 (line 200), the definition of the p-value p_{ij}^{c^k'} is unclear.  Does it represent the average of all p-values from the independence tests between X_i and X_j across all possible conditioning sets in c_k'?
* In Section 3.2.2 (lines 256-257), a brief description of "the rules described by Zhang" would be helpful. Specifically, it should clarify that these rules is about the orientation of edges in the causal graph.
* In Section 4.1 (line 288), the term "graph size" is vague. It should be explicitly defined as the number of nodes in the graph.
* The pseudocode for Algorithm 3 needs better formatting. The initial value of w_{ij}^{c^k'} is missing, and the first level of PSS is not included in the algorithm.

**Questions For Authors:**

1. How does the method perform when the probability distribution of samples varies across clients?
2. What is the rationale behind using the "union" operation to aggregate the local PAGs into a global graph?
3. The issues mentioned in the Theoretical Claims part need to be addressed. (see the Theoretical Claims part for details)
4. What recall results were obtained when evaluating the performance of the identified definite relationships?
5. What's the performance of FedCDnv compared to the method proposed in [1] by Wang et al. (2023)? (see the Essential References Not Discussed part for details)
6. Could you clarify the first two points mentioned in the Other Strengths and Weaknesses section?

A clear and detailed responses to question 1-5 would help.

**Relation To Broader Scientific Literature:**

The paper studies federated causal discovery when clients observe non-identical but overlapping variable sets, which is a novel and well-motivated problem.

The idea of using "stable relationship" to aggregate both "good" and "correct" relationships from local causal graphs to form a global causal graph is novel and effective.

The experimental results indicate that FedCDnv outperforms state-of-the-art methods across varying numbers of nodes and clients, demonstrating the effectiveness of the proposed method.

**Theoretical Claims:**

There are several issues with proofs of the theorem and lemmas in the paper:

Theorem 3.1: There is an issue with case (2.c). In the provided example, the set A_X^{G} should be {A, Y} rather than {A, B}. This error questions the correctness of this case.

Lemma 3.2: This lemma has several problems.
* First, the statement is unclear. It introduces Z_n as a set of "variable pairs" (like <X,Y>), yet later it states the "single node" X \in Z_n. This inconsistency needs to be addressed.
* Second, the proof is not well-explained.
  * In case (1), the learned casual relationship A->B alligns with the gound truth and should be definite. However, the lemma claims it is non-definite.
  * In case (2), the lemma says that X and Y "might" be adjacent, which suggests uncertainty, yet it does not clarify what happens if they are learned as non-causal.

Overall, the proof of Lemma 3.2 is not convincing.


Lemma 3.3: The proof seems to assume that X and Y are adjacent in the local graph, but this assumption is not mentioned in the lemma statement, which reduces clarity.

Lemma 3.4 appears to be correct.

---

> ### Author Rebuttal · Authors · 2025-04-01
>
> $\textbf{Responses for “Questions For Authors” are as follows.}$
>
> $\textbf{R1}$. Our method handles distributed data with varying samples distributions, where the observed variable sets are non-identical. We also experimentally evaluate the impact of $\delta$ on FedCDnv's performance. A lower $\delta$ indicates fewer overlapping variables and greater variability in the sample distribution. Fig.12 shows that as $\delta$ decreases from 95% to 60%, FedCDnv's performance drops by approximately 10%.
>
> $\textbf{R2}$. The "union" operation is applied only in the first aggregation phase, where adjacencies and arrowheads of all local graphs are transferred to a global graph. This global graph is then used to update the local graph in each client, without altering their skeletons and can even learn new orientations by using Zhang’s [2008 AI] rules (as illustrated in Fig. 3). The experiments in Figures 4-6 show a significant improvement in F1-orie.
>
> $\textbf{R3-1 for Theorem 3.1}$. Theorem 3.1 assumes that <X,Y> is a non-overlapping variable pair. Under case (2.c), if A o→ X ←o Y o→ B holds, then any dataset observing X will not observe Y, indicating that $A_X^{G}$ cannot be {A,Y}.
>
> $\textbf{R3-2 for Lemma 3.2 (First)}$. Sorry for this misunderstanding. $Z_n$ is the set of variables that appear in non-overlapping variable pairs. We will clarify it in the revised version.
>
> $\textbf{R3-2 for Lemma 3.2 (Second)}$.
>
> For case (1): Since the ground truth is unknown, we must consider both case(1) and case(2). When the condition in Lemma 3.2 is satisfied, either case (1) or (2) occurs, but we do not know which one occurs, which prevents us from guaranteeing the correctness of the learned relationships. Therefore, Lemma 3.2 claims it is non-definite.
>
> For case (2): Due to the unknown ground truth, as long as there exists any case where the learned relationship is non-definite, then the learned relationship is considered non-definite. Therefore, the situation that "they are learned as non-causal" no longer needs to be considered.
>
> $\textbf{R3-3 for Lemma 3.3}$. The proof does not assume that X and Y are adjacent in the local graph. It only assumes that X and Y are not adjacent in the ground truth. Here, the learned non-causal relationship between X and Y does not satisfy the condition "{A,B} $\subseteq \mathcal{A}_X^G \cup \mathcal{A}_Y^G$" in Lemma 3.3, meaning the premise does not hold, and thus it cannot be discussed in Lemma 3.3.
>
> $\textbf{R4}$. We use FDR to evaluate the reliability of FeddG by quantifying the proportion of false discoveries, where FeddG is the graph extracting only definite relationships from FedG. In contrast, recall is defined as True Positives / (True Positives + False Negatives). Since the denominator (True Positives + False Negatives) remains unchanged before and after extraction, the recall of FeddG (Rec-dC, Rec-dnC) is necessarily lower than (or equal with) that of FedG (Rec-C, Rec-nC). This is expected because FeddG is a subset of FedG, excluding non-definite relationships. The experimental results confirm this trend, as shown in the table below (Due to limited space, only a few are shown here).
> | nV  | Rec-C                | Rec-dC               | Rec-nC               | Rec-dnC              |
> |-----|----------------------|----------------------|----------------------|----------------------|
> | 20  | 0.60869 $\pm$ 0.20393 | 0.41304 $\pm$ 0.28344 | 0.99491 $\pm$ 0.01685 | 0.97344 $\pm$ 0.03836 |
> | 40  | 0.35476 $\pm$ 0.03688 | 0.22619 $\pm$ 0.02916 | 0.99567 $\pm$ 0.00638 | 0.98617 $\pm$ 0.00976 |
> | 60  | 0.54615 $\pm$ 0.10288 | 0.40769 $\pm$ 0.10383 | 0.99825 $\pm$ 0.00261 | 0.99200 $\pm$ 0.00784 |
> | 80  | 0.37650 $\pm$ 0.05856 | 0.23132 $\pm$ 0.07087 | 0.99868 $\pm$ 0.00154 | 0.98734 $\pm$ 0.00373 |
> | 100 | 0.58360 $\pm$ 0.04306 | 0.34098 $\pm$ 0.05933 | 0.99883 $\pm$ 0.00158 | 0.98761 $\pm$ 0.00652 |
>
> $\textbf{R5}$. Thanks for your comment, but there is no [1] that you mentioned.
>
> $\textbf{R6-1 for Weaknesses (lines 179-181)}$. Thanks for your comment. If the adjacency between $X_i$ and $X_j$ is not a bidirected edge but takes forms such as $X_i$ o—o $X_j$, $X_i$ o→ $X_j$ and $X_i$ ←o $X_j$, we provide a condition to address such cases. Specifically, if for every $G_{k_n} \subseteq$ {$G_{k_n}$}, there exists $Z \subseteq O_{k_n}$ such that $X⊥Y∣Z$ holds in $D_{k_n}$, and in every $G_{k_a}\in$ {$G_{k_a}$}, the variables in $Z$ are never observed simultaneously ($X_i,X_j \notin Z$), then the conflicting adjacency arises due to non-identical observed variable sets. We will clarify this in the revised version.
>
> $\textbf{R6-2 for Weaknesses (line 200)}$. If $X_i$ and $X_j$ are adjacent in $G_{k'}$, $p_{ij}^{c_k'}$ represents the average p-value from independence tests between $X_i$ and $X_j$ across all possible conditioning sets in $c_k'$. If $X_i$ and $X_j$ are non-adjacent in $G_{k'}$, $p_{ij}^{c_k'}$ corresponds to the p-value associated with the separating set that renders them independent.

---

### Official Review · Reviewer_YBLn · 2025-03-13

**Overall Recommendation:** 4

**Summary:**

This paper proposes novel algorithm FedCDnv, a federated method for learning causal structure where different clients observe non-identical variable sets. It mainly addresses two challenges: 1) spurious dependencies introduced by non-overlapping variable pairs, which may lead to incorrect causal conclusions, and 2) Varying importance of (non-)causal relationships between different variables within a client, requiring a careful aggregation mechanism. It also develops theories to detect spurious dependencies, defining stable relationships as those that are both "correct" and "good" across graphs discovered by multiple clients, and bridging the local learning with federated aggregation. The experiments conducted on synthetic, benchmark, and real-world datasets support the claims.

**Claims And Evidence:**

Yes. The paper provides theoretical and empirical support for the claims where the proof support the detecting spurious dependencies and the experiment includes comparisons with distributed and federated CSL methods, testing on synthetic, benchmark, and real-world datasets.
However, one potential weakness is that the paper does not deeply analyze or discuss the worst-case performance of FedCDnv, i.e., scenarios where FedCDnv underperforms or where its assumptions may not hold.

**Essential References Not Discussed:**

The paper could cite: (1) Alternative methods for handling latent confounders in federated learning, such as "Causal inference with latent variables: Recent advances and future prospectives." (2) Federated algorithms in other domains, such as "scFed: federated learning for cell type classification with scRNA-seq".

**Experimental Designs Or Analyses:**

Yes, the experimental setup is sound as it include distributed and federated CSL methods and several benchmarks. An improvement would be testing on more real-world datasets to enhance generalizability.

**Methods And Evaluation Criteria:**

Yes. The proposed methods and the evaluation criteria make sense for the problem of FCD with non-identical variable sets, including multiple benchmark in synthetic and real-world datasets for the specific problem.

**Other Comments Or Suggestions:**

- Lines 273-274 in Page 5 "… FedG as definite ones, obtaining FeddG". The notation could be clarified for better readability.
- The notation of $\textbf{Z}$ and $Z$ in Alg. 2 is somewhat confusing, as $Z$ generally refers to a variable within the set $\textbf{Z}$. Additionally, the presentation in line 4 of Alg. 2 could be improved for greater clarity. Line 198R: $\frac{n_k}{n}$ should be corrected as $\frac{n_{k’}}{n}$.
- The capitalization format of section titles is inconsistent. For example, Section 3.4 should be titled "Privacy and Costs Analysis" for consistency. It is recommended that the authors thoroughly review the manuscript to ensure uniform formatting.

**Other Strengths And Weaknesses:**

Strengths:
- The problem studied in this paper is both novel and practically significant. The challenges arising from non-identical variable sets across clients in federated settings are nontrivial and require careful consideration.
- The motivation for the study is clear and compelling, particularly regarding the spurious dependencies caused by non-overlapping variable pairs, which indeed require serious attention. (3) The underlying assumptions and theoretical proofs are well-established, and the experimental evaluation provided is thorough, covering two critical aspects comprehensively.

Weaknesses:
- Alg.3 seems a bit misleading. In line 9, $w_{ij}$ is calculated using Eq. (2), which requires scaled p-values. However, it is unclear where these scaled p-values are derived from, and no further detailed explanation is provided.
- Limited real-world dataset evaluation, with most results relying on benchmarks.
- There is no discussion of worst-case performance, making it unclear how FedCDnv behaves under extreme heterogeneity.

**Questions For Authors:**

1. Does the term “an oracle of conditional independence tests” refer to perfectly accurate CI tests?
2. Could the authors clarify the process of Alg.3?
3. How does the communication cost scale with an increasing number of clients? Have you explored optimizations for reducing overhead?
4. Can FedCDnv be extended to handle intervention data, given that CDUIOV explicitly models interventions?

**Relation To Broader Scientific Literature:**

This work proposes a federated method for learning causal structure which extend the prior work by allowing different clients observe non-identical variable sets, considering the presence of latent variables, detecting spurious dependencies, and computing the varying importance of (non-)causal relationships between different variables within a client. While prior FCD methods, such as FedPC, FedCSL, NOtears-ADMM, assume identical variable sets. Distributed methods, such as CDUIOV and CD-MiNi, ignore the might incorrect causal conclusions caused by non-overlapping variable pairs. Both makes FedCDnv a significant contribution to handling real-world non-identical variables.

**Theoretical Claims:**

Yes, the proofs for the proposed theories appear logically sound where Theorem 3.1 formalizes conditions under which non-overlapping variable for non-causal relationships, Lemmas 3.2-3.3 formalize criteria for detecting spurious dependencies. Lemma 3.4 provides a heuristic rule for identifying unobserved confounders, though it might be better labeled as a "proposition" to reflect its heuristic nature.

---

> ### Author Rebuttal · Authors · 2025-04-01
>
> $\textbf{Responses for “Questions For Authors” are as follows.}$
>
> $\textbf{Q1}$. Does the term “an oracle of conditional independence tests” refer to perfectly accurate CI tests?
>
> $\textbf{R1}$. Yes, “an oracle of conditional independence tests” refers to fully accurate CI tests.
>
> $\textbf{Q2}$. Could the authors clarify the process of Alg.3?
>
> $\textbf{R2}$. Algorithm 3 is proposed for implementing the designed two-level priority selection strategy (PSS). The inputs include the updated local causal graph Pag$_u^{c_k}$, sample size $n_k$, and stability matrix R$_s^{c_k}$ of each client, while the output is the global causal graph FedG over the integrated variable set $O$.
>
> During aggregation, adjacencies are determined by comparing $val_{ij}$ (for $X_i$ and $X_j$) with 0. The computation of $val_{ij}$ depends on $w_{ij}^{c_k}$ and the sample size weight $w_{c_k}$. Specifically, if the relationship between $X_i$ and $X_j$ is stable, $w_{ij}^{c_k}$ is based solely on $w_{c_k}$. Otherwise, it is derived from the product of $w_{c_k}$ and the $w_{ij}$ value computed by each client (as Eq. (2)). Stability is encoded in the stability matrix, where each entry represents the stability score (2, -2, 1, -1) multiplied by the scaled p-value. Consequently, $w_{ij}$ is computed by dividing each entry by the corresponding stability score.
>
> For orientation aggregation, only arrows are considered. If any local graph contains an arrow X→Y, it is incorporated into FedG as X $\circ\hspace{-0.43em}\rightarrow$ Y.
>
> $\textbf{Q3}$. How does the communication cost scale with an increasing number of clients? Have you explored optimizations for reducing overhead?
>
> $\textbf{R3}$. We have theoretically analyzed the communication cost of the proposed FedCDnv algorithm, which is $O(4md^2)$ and increases linearly with the number of clients. We are aware that the stability matrix R$_s^{c_k}$ contributes to communication costs. In future work, we will explore optimizations to reduce this overhead.
>
> $\textbf{Q4}$. Can FedCDnv be extended to handle intervention data, given that CDUIOV explicitly models interventions?
>
> $\textbf{R4}$. Yes, FedCDnv can be extended to handle interventional data. CDUIOV learns causal structures over the integrated set of variables from interventional datasets across multiple domains. It assumes that within each domain, interventions are performed on an identical set of variables. CDUIOV aims to address inconsistencies in causal relationships caused by unknown intervention targets and non-overlapping variable pairs. Therefore, when each client holds multiple interventional datasets, FedCDnv can integrate these local structures into a global causal graph while preserving data privacy.
>
> $\textbf{Responses for “Weaknesses” are as follows.}$
>
> $\textbf{Q5-1}$. Alg.3 seems a bit misleading.
>
> $\textbf{R5-1}$. Please see R2.
>
> $\textbf{Q5-2}$. Limited real-world dataset evaluation, with most results relying on benchmarks.
>
> $\textbf{R5-2}$. We have conducted experiments using the real-world Sachs dataset, which consists of measurements from 11 phosphorylated proteins and phospholipids in individual cells. The experimental results are presented in Table 1 and Table 6.
> In addition, this study also has real-world applications. We found the eICU Collaborative Research Database (eICU-CRD)  (link https://physionet.org/content/eicu-crd/2.0/), a real-world dataset available on the PhysioNet website, as a motivating example for applying FedCDnv (See R2 of Reviewer 2 for details).
>
> $\textbf{Q5-3}$. There is no discussion of worst-case performance, making it unclear how FedCDnv behaves under extreme heterogeneity.
>
> $\textbf{R5-3}$. In the experimental section, we analyzed the worst-case performance of FedCDnv under increasing heterogeneity. When the observed variables differ across clients, the data distribution of each client also varies. We experimentally evaluated the impact of different values of $\delta$ on FedCDnv's performance, where a lower $\delta$ indicates fewer overlapping variables and greater variation in sample distributions. Fig. 12 shows that as $\delta$  decreases from 95% to 60%, FedCDnv's performance drops by approximately 10%.

---

### Official Review · Reviewer_od7H · 2025-03-13

**Overall Recommendation:** 4

**Summary:**

The paper introduces FedCDnv, a federated causal structure learning algorithm designed for scenarios where clients have non-identical but overlapping variable sets. The method introduces theoretical criteria to distinguish definite causal and non-causal relationships. A two-level priority selection strategy (PSS) is developed to aggregate both “correct” (definite) and “good” (stable) relationships from local causal graphs into a global causal graph.

**Claims And Evidence:**

Yes, the claims (detecting good and correct causal relationships) are backed up by theoretical statements with proofs.

**Essential References Not Discussed:**

None, to the best of my knowledge.

**Experimental Designs Or Analyses:**

Yes, all the experiments in the main text and the parameter-sensitivity experiments in Appendix D.4.

**Methods And Evaluation Criteria:**

Yes, standard causal discovery metrics (e.g., F1) are used for gauging the accuracy of the learned graph; and the datasets used (synthetic and real) make sense for the problem.

**Other Comments Or Suggestions:**

1. Figure 1 is never mentioned/referenced in the text.
2. Line 294 Section 4.1 should read "real-**world** data."
3. The error bars in the figures are hardly visible. I suggest either using error bands, or also displaying the whiskers (lines).
4. Tables 1 and 2 contain too many significant digits to be easily legible. I suggest reporting figures to 2 or max 3 decimal places.

**Other Strengths And Weaknesses:**

**Strengths**
1. The problem of FCD with non-overlapping variables is novel and realistic one in practice.
2. Theoretical justifications are provided (e.g.,correctness of identifying definite causal and non-causal relationships).
3. Privacy and communication costs are given.

**Weaknesses**
1. The communication costs of a client $c_k$ scale with $O(d_k^2)$, where $d_k$ is the number of variables observed by the client.

**Questions For Authors:**

1. Could the authors provide a real-world motivating example where FCD is used/needed and there are non-overlapping variables observed by clients?

**Relation To Broader Scientific Literature:**

The work extends federated causal discovery to non-identical variable sets, addressing gaps in prior works (e.g., FedCSL, Notears-ADMM) that assume identical variables.

**Theoretical Claims:**

Yes, Theorem C.1. No issues were found.

---

> ### Author Rebuttal · Authors · 2025-04-01
>
> $\textbf{Responses for “Questions For Authors" are as follows.}$
>
> $\textbf{Q1}$. Could the authors provide a real-world motivating example where FCD is used/needed and there are non-overlapping variables observed by clients?
>
> $\textbf{R1}$. Thanks for your comment. A real-world example where non-overlapping variables are observed can be found in the eICU Collaborative Research Database (eICU-CRD, link https://physionet.org/content/eicu-crd/2.0/) [1-3]. It is a multi-center intensive care unit database covering over 200,000 admissions to ICUs across the United States between 2014-2015. In the "vitalPeriodic.csv" file, different ICU stays (identified by "patientunitstayid") record partially overlapping physiological variables, as shown below:
> | vital-periodic-id | patient-unit-stay-id | sao2 | heart-rate | respiration | cvp | system-ic-systolic | system-ic-diastolic | system-ic-mean | pa-systolic | pa-diastolic | pa-mean | icp-st1 | icp-st2 | icp-st3 |
> |-------------------|----------------------|------|------------|-------------|-----|--------------------|---------------------|----------------|-------------|--------------|---------|---------|---------|---------|
> | 35511110          | 141945               | 100  | 76         | 17          |     |                    |                     |                |             |              |         | -1      | -0.81   | 0       |
> | 35417390          | 141945               | 89   | 87         | 24          |     |                    |                     |                |             |              |         | -1      | -0.69   | 0.1     |
> | 28818781          | 142000               | 96   | 108        | 30          | 8   | 134                | 48                  | 74             |             |              |         |         |         |         |
> | 28839535          | 142000               | 98   | 80         | 20          | 11  | 142                | 50                  | 82             |             |              |         |         |         |         |
> | 48431259          | 142035               | 98   | 98         | 29          | 17  | 116                | 60                  | 76             | 34          | 13           | 23      |         |         |         |
> | 48435712          | 142035               | 94   | 92         | 23          | 16  | 116                | 68                  | 82             | 34          | 14           | 25      |         |         |         |
>
> In general, clinical practice requires access to up-to-date patient data from hospitals, which is often distributed and privacy-sensitive. Due to regulatory constraints such as HIPAA and institutional policies, hospitals cannot directly share raw and up-to-date patient data for collaborative analysis. This creates a critical need for federated causal structure learning with non-identical variables sets, which allows clients to collaboratively infer causal relationships over integrated variables while keeping data decentralized and secure.
>
> [1] Pollard, T., Johnson, A., Raffa, J., Celi, L. A., Badawi, O., & Mark, R. (2019). eICU Collaborative Research Database (version 2.0). PhysioNet. https://doi.org/10.13026/C2WM1R.
>
> [2] Nakayama LF, Restrepo D, Matos J, Ribeiro LZ, Malerbi FK, Celi LA, Regatieri CS. BRSET: A Brazilian Multilabel Ophthalmological Dataset of Retina Fundus Photos. PLOS Digit Health. 2024 Jul 11;3(7):e0000454. doi: 10.1371/journal.pdig.0000454. PMID: 38991014; PMCID: PMC11239107.
>
> [3] Goldberger, A., Amaral, L., Glass, L., Hausdorff, J., Ivanov, P. C., Mark, R., ... & Stanley, H. E. (2000). PhysioBank, PhysioToolkit, and PhysioNet: Components of a new research resource for complex physiologic signals. Circulation [Online]. 101 (23), pp. e215–e220.
>
> $\textbf{Responses for “Other Comments Or Suggestions" are as follows.}$
>
> $\textbf{Q2-1}$. Figure 1 is never mentioned/referenced in the text.
>
> $\textbf{R2-1}$. Thank you for pointing this out. We apologize for the oversight. Figure 1 is used in the fourth paragraph of the introduction to illustrate spurious dependencies, but it was not explicitly referenced in the text. In the revision, we will embed Figure 1 within the fourth paragraph of the introduction.
>
> $\textbf{Q2-2}$. Line 294 Section 4.1 should read "real-world data."
> The error bars in the figures are hardly visible. I suggest either using error bands, or also displaying the whiskers (lines).
> Tables 1 and 2 contain too many significant digits to be easily legible. I suggest reporting figures to 2 or max 3 decimal places.
>
> $\textbf{R2-2}$. Thanks for your suggestions. We revised the errors and carefully checked the manuscript. In addition, for the error bars, we replaced them with whiskers (lines) to make them clearer. For Tables 1, 2 and 6, we retained three decimal places and revised them accordingly. Thanks again.

---

> > ### Comment · Reviewer_od7H · 2025-04-04
> >
> > I thank the authors for addressing my questions, and in particular for providing a real-world motivating example. I have increased my score.

---

> > > ### Author Response · Authors · 2025-04-04
> > >
> > > Thank you for your thoughtful review and for increasing your score. We sincerely appreciate your positive feedback and support.

---

### Official Review · Reviewer_7pKi · 2025-03-14

**Overall Recommendation:** 4

**Summary:**

This paper investigates federated causal structure learning, aiming to discover causal relationships between variables from data distributed across individual clients while considering privacy concerns. The paper addresses federated causal structure learning with non-identical variable sets and designs an effective strategy to aggregate "correct" and "good" relationships between variables during collaborative learning. Experimental results demonstrate that the proposed method is effective on synthetic, benchmark, and real-world data.

**Claims And Evidence:**

Yes. The claims are clear and convincing.

**Essential References Not Discussed:**

Yes. The related works are cited and discussed.

**Experimental Designs Or Analyses:**

Yes. I have checked the soundness of all the experimental designs and the experimental analyses are valid.

**Methods And Evaluation Criteria:**

Yes. The proposed methods and evaluation criteria make sense for the problem of federated causal structure learning with non-identical variable sets.

**Other Comments Or Suggestions:**

N/A

**Other Strengths And Weaknesses:**

**Strengths:**

1. This paper proposes a federated causal structure learning method to address the challenge of discovering causal relationships under non-identical variable sets.
2. The effectiveness of the method is validated through experiments on synthetic data, benchmark data, and real-world data, demonstrating its applicability and robustness across various types of data.

**Weaknesses:**

1. Theorems 3.2 and 3.3 only indicate that the relationship between variables X and Y is uncertain, but do not analyze under what conditions their relationship can be determined.
2. How does the proposed method aggregate the results obtained from different servers? The paper discusses taking the union of variable sets and the union of edge sets. When the relationships between two variables differ, such as Xo-oY, Xo->Y, and X->Y, what does the union of edges look like?

**Questions For Authors:**

See the weaknesses above.

**Relation To Broader Scientific Literature:**

The key contributions of the paper related to the problem of causal discovery with non-identical variable sets.

**Theoretical Claims:**

Yes. I have carefully reviewed all the proofs.

---

> ### Author Rebuttal · Authors · 2025-04-01
>
> $\textbf{Q1}$. Theorems 3.2 and 3.3 only indicate that the relationship between variables X and Y is uncertain, but do not analyze under what conditions their relationship can be determined.
>
> $\textbf{R1}$. Thanks very much for your comment. As stated in line 114 of the manuscript, we initially assume that all relationships among overlapping variable pairs are definite (or "correct"). Theorems 3.2 and 3.3 are proposed to detect which of these relationships become non-definite due to the influence of non-overlapping variable pairs. Therefore, the relationships that satisfy the conditions of Theorems 3.2 and 3.3 are non-definite, while all other relationships remain definite.
>
> $\textbf{Q2-1}$. How does the proposed method aggregate the results obtained from different servers?
>
> $\textbf{R2-1}$. Thanks for your comment. There are two aggregation stages.
>
> (1) For the first stage, the server integrates the adjacencies and orientations of local causal graphs learned by all clients into a global graph. For adjacencies, a variable pair is considered adjacent if any client reports the adjacency between them. For orientations (there are three types of orientations: circle 'o', tail '-', and arrow '>'), only the identified arrows are included in the global graph.
>
> (2) For the second stage, the server applies the proposed two-level priority selection strategy to aggregate all updated local graphs. First, for adjacency aggregation: a) First Priority Level: Check whether adjacency conflicts arise due to non-overlapping variable pairs, using Lemma 3.4; b) Second Priority Level: If the conflict is due to statistical errors, the final adjacency is determined by comparing $val_{ij}$ (for $X_i$ and $X_j$) with 0. The computation of $val_{ij}$ depends on two factors: $w_{ij}^{c_k}$ and the sample size weight $w_{c_k}$. Second, for orientation aggregation, the process is similar to stage (1).
>
> $\textbf{Q2-2}$. The paper discusses taking the union of variable sets and the union of edge sets. When the relationships between two variables differ, such as X $\\circ\hspace{-0.4em}-\hspace{-0.4em}\circ\$ Y, X $\circ\hspace{-0.43em}\rightarrow$ Y, and X$\rightarrow$Y, what does the union of edges look like?
>
> $\textbf{R2-2}$. When the relationships between two variables differ across clients, only identified arrows are included. For example, for X $\\circ\hspace{-0.4em}-\hspace{-0.4em}\circ\$ Y, X $\circ\hspace{-0.43em}\rightarrow$ Y, and X$\rightarrow$Y, the final relationship between X and Y will be X $\circ\hspace{-0.43em}\rightarrow$ Y.

---

> > ### Comment · Reviewer_7pKi · 2025-04-05
> >
> > I thank the author for the comprehensive response, particularly the detailed explanation of aggregating multiple servers results into a unified result. The clarifications have effectively addressed all my concerns, and I have increased my score.

---

> > > ### Author Response · Authors · 2025-04-05
> > >
> > > Thank you very much for your thoughtful review. We sincerely appreciate your positive comments and the increased score.

---

### Decision · Program_Chairs · 2025-05-01

**Decision:**

Accept (poster)

**Comment:**

This paper addresses the problem of federated causal discovery with non-identical but overlapping variable sets, extending the scope of existing federated methods that typically assume identical variables across clients. All reviewers found the problem setting timely, and the methodological contributions reasonable. One reviewer raised concerns about the clarity and rigor of certain theoretical arguments but acknowledged the conceptual soundness of the approach. The authors provided clarifications in the rebuttal, which led to improved scores from other reviewers. Overall, the paper makes a meaningful contribution to federated causal discovery under more realistic settings, and I recommend acceptance.